# Task-agnostic Pre-training and Task-guided Fine-tuning for Versatile Diffusion Planner

## Abstract

Diffusion models have demonstrated their capabilities in modeling trajectories of multi-tasks. However, existing multi-task planners or policies typically rely on task-specific demonstrations via multi-task imitation, or require task-specific reward labels to facilitate policy optimization via Reinforcement Learning (RL). They heavily rely on task-specific labeled data, which can be difficult to acquire. To address these challenges, we aim to develop a versatile diffusion planner that can leverage large-scale inferior data that contains task-agnostic sub-optimal trajectories, with the ability to fast adapt to specific tasks. In this paper, we propose **SODP**, a two-stage framework that leverages **S**ub-**O**ptimal data to learn a **D**iffusion **P**lanner, which is generalizable for various downstream tasks. Specifically, in the pre-training stage, we train a foundation diffusion planner that extracts general planning capabilities by modeling the versatile distribution of multi-task trajectories, which can be sub-optimal and has wide data coverage. Then for downstream tasks, we adopt RL-based fine-tuning with task-specific rewards to quickly refine the diffusion planner, which aims to generate action sequences with higher task-specific returns. Experimental results from multi-task domains including Meta-World and Adroit demonstrate that SODP outperforms state-of-the-art methods with only a small amount of data for reward-guided fine-tuning.

## 1 Introduction

There has been a long-standing pursuit to develop agents capable of performing multiple tasks (Reed et al., 2022; Lee et al., 2022). Although traditional RL methods have made significant strides in training agents to master individual tasks (Silver et al., 2016; OpenAI et al., 2019), expanding this capability to handle diverse tasks remains a significant challenge due to the diversity of task variants and optimization directions with different rewards. Multi-task RL aims to address this by developing agents via task-conditioned optimization (Yu et al., 2020; Lee et al., 2022) or parameter-compositional learning (Sun et al., 2022; Lee et al., 2023). However, the inherent diversity in task trajectory distributions makes it challenging to model and accommodate modeling across different task structures. Diffusion models (Sohl-Dickstein et al., 2015; Ho et al., 2020), originally designed for generative tasks, provide a powerful framework to address these difficulties. Their capacity to capture complex, multi-modal distributions within high-dimensional data spaces (Podell et al., 2023; Ho et al., 2022; Jing et al., 2022) makes them well suited to represent the broad variability encountered in multi-task environments.

Motivated by this, existing methods have employed diffusion models to mimic expert behaviors derived from human demonstrations on various tasks (Pearce et al., 2023; Xu et al., 2023; Chi et al., 2023). However, acquiring task-specific demonstrations is often time-consuming and costly, especially in environments requiring specialized domain expertise. Alternative approaches attempt to optimize diffusion models with return guidance (He et al., 2024; Liang et al., 2023) or conventional RL paradigm (Wang et al., 2022b), which demands a large volume of data with reward labels for each task. To address the above limitations, we wonder whether a generalized diffusion planner can be learned from a large amount of low-quality trajectories without reward labels, with the ability to adapt quickly to various downstream tasks. We only require the inferior data to comprise a mixture of sub-optimal state-action pairs from various tasks, which can be easily obtained in the real world. In training, the diffusion planner seeks to model the distribution of diverse trajectories with broad coverage, enabling it to acquire generalizable capabilities and allowing the planner to further con-

**Offline multi-task actions**  **Online task-specific actions**

(a) Stage 1: Pre-training on multiple task datasets  (b) Stage 2: Fine-tuning on each specific task

Figure 1: The Overall framework. Different colors represent different tasks. The diffusion model is first pre-trained on a mixed dataset drawn from multiple tasks, and is then fine-tuned for each specific task using task-specific rewards.

centrate on high-reward regions of specific downstream tasks via fast adaptation. An overview of our method is given in Figure 1.

In this paper, we propose a novel framework to utilize **S**ub-**O**ptimal data to train a **D**iffusion **P**lanner (**SODP**) that can generalize across a wide range of downstream tasks. SODP consists of two stages: pre-training and fine-tuning. By leveraging a set of trajectories of different tasks for pre-training, we employ action-sequence prediction to capture shared knowledge across tasks. Since the state space may vary between tasks, focusing on the common action space (e.g., end-effector poses of a robot arm) facilitates task generalization. We frame the pre-training stage as a conditional generative problem that generates future actions based on historical states. Then, inspired by the remarkable success of RL-based alignment for LLMs (Ouyang et al., 2022; Glaese et al., 2022), we adopt an RL-based fine-tuning approach to tailor the pre-trained diffusion planner to specific downstream tasks. Specifically, we conduct online interaction based on the pre-trained planner to collect task-specific experiences with reward labels, and perform policy gradients to iteratively refine the predicted action-sequence distribution based on reward feedback of tasks. Through fine-tuning, the diffusion planner can gradually adapt toward generating actions with high task-specific rewards and eventually become optimal for the given task.

Figure 2 illustrates our method. In pre-training, the model captures diverse behavior patterns from the training data, encompassing inferior and mediocre actions. After fine-tuning, the model shrinks the action distribution and concentrates on generating optimal action sequences for a specific task. Our contributions can be summarized as follows. (i) We propose a novel pre-training and fine-tuning paradigm for learning a versatile diffusion planner, which leverages sub-optimal transitions to capture the broad action distributions across tasks, and adopt task-specific fine-tuning to transfer the planner to downstream tasks. (ii) We give an efficient fine-tuning algorithm based on policy gradient for diffusion planners, which progressively shifts the action distribution to concentrate on re-

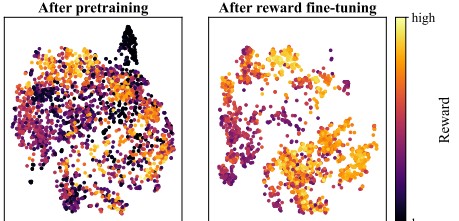

Figure 2: Illustration of SODP in Meta-World *button-press-wall* task. We present trajectories generated by the diffusion model after pre-training and fine-tuning of SODP. The pre-trained model captures a wide range of behaviors, and the fine-tuned model discards the inferior behaviors to coverage to high-reward regions.

gions associated with higher task returns. (iii) We conduct extensive experiments using sub-optimal data from state-based Meta-World (Yu et al., 2019) as well as image-based Adroit (Rajeswaran et al., 2017), showcasing its superior performance compared to state-of-the-art approaches.

## 2 PRELIMINARIES

**Multi-task RL** We consider the multi-task RL problem involving $N$ tasks, where each task $\mathcal{T} \sim p(\mathcal{T})$ is represented by a task-specified Markov Decision Process (MDP). Each MDP is defined by a

tuple $(\mathcal{S}^{\mathcal{T}}, \mathcal{A}, P^{\mathcal{T}}, R^{\mathcal{T}}, P_0^{\mathcal{T}}, \gamma)$, where $\mathcal{S}^{\mathcal{T}}$ is the state space of task $\mathcal{T}$, $\mathcal{A}$ is the global action space, $P^{\mathcal{T}}(s_{t+1}^{\mathcal{T}}|s_t^{\mathcal{T}}, a_t^{\mathcal{T}}) : \mathcal{S}^{\mathcal{T}} \times \mathcal{A} \to \mathcal{S}^{\mathcal{T}}$ is the transition function, $R^{\mathcal{T}}(s_t^{\mathcal{T}}, a_t^{\mathcal{T}}) : \mathcal{S}^{\mathcal{T}} \times \mathcal{A} \to \mathbb{R}$ is the reward function, $\gamma \in (0, 1]$ is the discount factor, and $P_0^{\mathcal{T}}$ is the initial state distribution. We assume that all tasks share a common action space, executed by the same agent, while differing in their respective reward functions, state spaces, and transition dynamics. At each timestep $t$, the agent perceives a state $s_t^{\mathcal{T}} \in \mathcal{S}^{\mathcal{T}}$, takes an action $a_t^{\mathcal{T}} \in \mathcal{A}$ according to the policy $\pi^{\mathcal{T}}(a_t^{\mathcal{T}}|s_t^{\mathcal{T}})$, and receives a reward $r_t^{\mathcal{T}}$. The agent's objective is to determine an optimal policy that maximizes the expected return across all tasks: $\pi^* = \arg\max_\pi \mathbb{E}_{\mathcal{T} \sim p(\mathcal{T})} \mathbb{E}_{a_t \sim \pi^{\mathcal{T}}} \left[ \sum_{t=0}^{\infty} \gamma^t r_t^{\mathcal{T}} \right]$.

**Diffusion Models**  Diffusion models (Sohl-Dickstein et al., 2015) are a type of generative model that first add noise to the data $\boldsymbol{x}_0$ from a unknown distribution $q(\boldsymbol{x}_0)$ in $K$ steps through a forward process defined as:

$$q(\boldsymbol{x}_k|\boldsymbol{x}_{k-1}) := \mathcal{N}(\boldsymbol{x}_k; \sqrt{1 - \beta_k}\boldsymbol{x}_{k-1}, \beta_k \boldsymbol{I}), \tag{1}$$

where $\beta_k$ is a predefined variance schedule. Then, a trainable reverse process is constructed as:

$$p_\theta(\boldsymbol{x}_{k-1}|\boldsymbol{x}_k) := \mathcal{N}(\boldsymbol{x}_{k-1}; \mu_\theta(\boldsymbol{x}_k, k), \Sigma_k), \tag{2}$$

where $\mu_\theta(\boldsymbol{x}_k, k)$ is the forward process posterior mean as a function of a noise prediction neural network $\epsilon_\theta(\boldsymbol{x}_k, k)$ with a learnable parameter $\theta$ (Ho et al., 2020). $\epsilon_\theta(\boldsymbol{x}_k, k)$ can be trained via a surrogate loss as

$$\mathcal{L}_{\text{denoise}}(\theta) := \mathbb{E}_{k \sim [1,K], x_0 \sim q, \epsilon \sim \mathcal{N}(0, \boldsymbol{I})} \left[ \left\| \epsilon - \epsilon_\theta(\boldsymbol{x}_k, k) \right\|^2 \right]. \tag{3}$$

After training, samples can be generated by first drawing Gaussian noise $\boldsymbol{x}_K$ and then iteratively denoising $\boldsymbol{x}_K$ into a noise-free output $x_0$ over $K$ iterations using the trained model $\epsilon_\theta(\boldsymbol{x}_k, k)$ by

$$\boldsymbol{x}_{k-1} = \frac{1}{\sqrt{\alpha_k}} \left( \boldsymbol{x}_k - \frac{1 - \alpha_k}{\sqrt{1 - \bar{\alpha}_k}} \epsilon_\theta(\boldsymbol{x}_k, k) \right) + \sigma_k \mathcal{N}(0, \boldsymbol{I}), \tag{4}$$

where $\alpha_k := 1 - \beta_k$, $\bar{\alpha}_k := \prod_{s=1}^{k} \alpha_s$ and $\sigma_k = \sqrt{\beta_k}$.

## 3 METHOD

We propose SODP, a two-stage framework that leverages large amounts of sub-optimal data to train a diffusion planner that can generalize to downstream tasks. The process is depicted in Figure 3. In the pre-training stage, we train a guidance-free diffusion model to predict future actions based on historical states, using an mixture offline dataset cross tasks without reward labels. In the fine-tuning stage, we refine the pre-trained model using policy gradient to maximize the task-specific rewards, additionally incorporating a regularization term to prevent the model from losing acquired skills.

### 3.1 PRE-TRAINING WITH LARGE-SCALE SUB-OPTIMAL DATA

Previous works (He et al., 2024) typically model multi-task RL as a conditional generative problem using diffusion models trained on datasets composed of multiple task subsets $\mathcal{D} = \cup_{i=1}^{N} \mathcal{D}_i$, as:

$$\max_\theta \mathbb{E}_{\tau \sim \cup_i \mathcal{D}_i} \left[ \log p_\theta(\boldsymbol{x}_0(\tau) \mid \boldsymbol{y}(\tau) \right], \tag{5}$$

which requires additional condition $\boldsymbol{y}(\tau)$ to guide diffusion model to generate desirable trajectories. For instance, $\boldsymbol{y}(\tau)$ should contain the return of trajectory $R(\tau)$ and task description $Z$ as prompt. However, the reward label and trajectory description may be scarce or costly to obtain in the real-world. To overcome this challenge, we train a diffusion planner that can learn from offline trajectories transitions (i.e., $\{(s_t, a_t, s_{t+1})\}$) without reward label or task descriptions. Specifically, we model the problem as a guidance-free generation process (Chi et al., 2023):

$$\max_\theta \mathbb{E}_{(\boldsymbol{s}_t, \boldsymbol{a}_t) \sim \cup_i \mathcal{D}_i} \left[ \log p_\theta(\boldsymbol{a}_t^0 \mid \boldsymbol{s}_t) \right]. \tag{6}$$

Here, we represent $\boldsymbol{x}_0 := \boldsymbol{a}_t^0 = (a_t, a_{t+1}, ..., a_{t+H-1})$ as an action sequence, where $H$ is the planning horizon and $t$ is the timestep sampled from dataset $\mathcal{D}$. As previous work (Chi et al., 2023), we denote $\boldsymbol{s}_t$ as the historical states at timestep $t$ with length $T_o$, i.e., $\boldsymbol{s}_t := \{s_{t-T_o+1}, \ldots s_{t-1}, s_t\}$. The formulation in Eq. (6) enables the model to learn the broad action-sequence distribution of

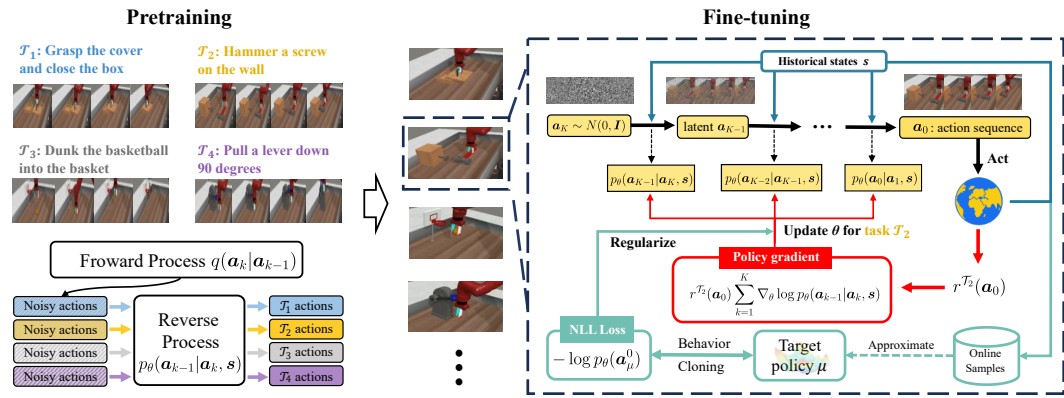

Figure 3: Overview of SODP. We initially pre-train a diffusion model using multi-task transition data to predict action sequences from historical states. Subsequently, we fine-tune the model on downstream tasks using policy gradient methods, incorporating a regularization term to mitigate model degradation.

multi-tasks depending on previous observations, without requiring additional guidance. To train our planning model, we modify Eq. (3) to obtain our pre-training objective as follows:

$$\mathcal{L}_{\text{pre-train}}(\theta) = \mathbb{E}_{k\sim[1,K],(\boldsymbol{s}_t,\boldsymbol{a}_t^0)\sim D,\epsilon\sim\mathcal{N}(0,\mathbf{I})}\left[\left\|\epsilon - \epsilon_\theta(\boldsymbol{a}_t^k,\boldsymbol{s}_t,k)\right\|^2\right]. \tag{7}$$

Following Eq. (4), we can generate action sequences through a series of denoising steps:

$$\boldsymbol{a}_t^{k-1} = \frac{1}{\sqrt{\alpha_k}}\left(\boldsymbol{a}_t^k - \frac{1-\alpha_k}{\sqrt{1-\bar{\alpha}_k}}\epsilon_\theta(\boldsymbol{a}_t^k,\boldsymbol{s}_t,k)\right) + \sigma_k\mathcal{N}(0,\boldsymbol{I}). \tag{8}$$

Unlike other models, the dataset $\mathcal{D}$ we used for the pre-training stage is not restricted to expert trajectories. As shown in Figure 2, we aim to train a foundation model that captures diverse behaviors and learns general capabilities from inferior trajectories, enabling the planner to enhance its representation and action priors through pre-training before learning on downstream tasks.

## 3.2 Reward Fine-tuning for Downstream Tasks

**MDP notation.** The fine-tuning stage involves two distinct MDPs: one for RL decision process and the other for the diffusion model denoising process. We use the superscript diff (e.g., $s_k^{\text{diff}}$, $a_k^{\text{diff}}$) to denote the MDP associated with diffusion model denoising process, while no superscript is used for the MDP related to the RL process (e.g., $s_t, a_t$). Additionally, we use $k \in \{K, \dots, 0\}$ to represent the diffusion timestep and $t \in \{1, \dots, T\}$ to represent the trajectory timestep.

We model the denoising process of our pre-trained diffusion planner as a $K$-step MDP as follows:

$$s_k^{\text{diff}} = (s_t, \boldsymbol{a}_t^{K-k}), \qquad a_k^{\text{diff}} = \boldsymbol{a}_t^{K-k-1}, \qquad P_0^{\text{diff}}(s_0^{\text{diff}}) = (\delta_{s_t}, \mathcal{N}(0, I)),$$

$$P^{\text{diff}}(s_{k+1}^{\text{diff}} \mid s_k^{\text{diff}}, a_k^{\text{diff}}) = (\delta_{s_t}, \delta_{a_k^{\text{diff}}}), \qquad R^{\text{diff}}(s_k^{\text{diff}}, a_k^{\text{diff}}) = \begin{cases} r(s_{k+1}^{\text{diff}}) = r(\boldsymbol{a}_t^0) & \text{if } k = K-1, \\ 0 & \text{otherwise.} \end{cases},$$

$$\pi_\theta^{\text{diff}}(a_k^{\text{diff}} \mid s_k^{\text{diff}}) = p_\theta(\boldsymbol{a}_t^{K-k-1} \mid \boldsymbol{a}_t^{K-k}, \boldsymbol{s}_t), \tag{9}$$

where $s_k^{\text{diff}}$ and $a_k^{\text{diff}}$ are the state and action at timestep $k$, $P_0^{\text{diff}}$ and $P^{\text{diff}}$ are the initial distribution and transition dynamics, $\delta$ is the Dirac delta distribution, $R^{\text{diff}}$ is the reward function and $p_\theta(\boldsymbol{a}_t^{K-k-1} \mid \boldsymbol{a}_t^{K-k}, \boldsymbol{s}_t)$ is the pre-trained diffusion planner. This formulation allows the state transitions in the MDP to be mapped to the denoising process in the diffusion model. The MDP initiates by sampling an initial state $s_0^{\text{diff}} \sim P_0^{\text{diff}}$, which corresponds to sample Gaussian noise $\boldsymbol{a}_t^K$ at the beginning of the reverse process. At each timestep $k$, the policy $\pi_\theta^{\text{diff}}(a_k^{\text{diff}} \mid s_k^{\text{diff}})$ takes an action $a_k^{\text{diff}}$ based on current state $s_k^{\text{diff}}$, which corresponds to generate next latent $\boldsymbol{a}_t^{K-k-1}$ based on current latent $\boldsymbol{a}_t^{K-k}$ following Eq. (8). The reward remains zero until a noise-free output $\boldsymbol{a}_t^0$ is evaluated. Different from

previous text-to-image studies that typically evaluate the final sample using a pre-trained reward model (Black et al., 2023; Fan et al., 2024), we aim to fine-tune the pre-trained diffusion planner to maximize rewards of downstream tasks, which makes constructing reward models for all tasks costly. Therefore, we directly evaluate the generated action sequences in an online RL environment for each specific task $\mathcal{T}$. Specifically, for any given timestep $t$, we use the planner to generate future actions $\boldsymbol{a}_t^0 = (a_t, a_{t+1}, \ldots, a_{t+H-1})$ and then execute the first $T_a$ steps. Then we calculate the discounted cumulative reward from the environment to assess the generated sample, expressed as $r(\boldsymbol{a}_t^0) = \sum_t^{T_a} \gamma^{t-1} r^{\mathcal{T}}(s_t, a_t)$. We write $r(\boldsymbol{a}_t^0)$ as shorthand for $r(s_t, \boldsymbol{a}_t^0)$ for brevity.

**Finetuning objective.** The objective of fine-tuning our pre-trained diffusion planner is to maximize the expected reward of the generated action sequences for the target downstream task $\mathcal{T}$, which can be defined as:

$$J^{\mathcal{T}}(\theta) = \sum_t \mathbb{E}_{p_\theta(\boldsymbol{a}_t^0 | \boldsymbol{s}_t)}[r^{\mathcal{T}}(\boldsymbol{a}_t^0)]. \tag{10}$$

Directly optimizing the objective $J^{\mathcal{T}}(\theta)$ is intractable since it is infeasible to evaluate the return over all possible actions. Therefore, we utilize policy gradient methods (Sutton et al., 1999), which estimate the policy gradient and apply a stochastic gradient ascent algorithm for updates. The gradient of the objective $J^{\mathcal{T}}(\theta)$ can be obtained as follows:

$$\nabla_\theta J^{\mathcal{T}}(\theta) = \sum_t \mathbb{E}_{p_\theta(\boldsymbol{a}_t^{0:K} | \boldsymbol{s}_t)} \left[ r^{\mathcal{T}}(\boldsymbol{a}_t^0) \sum_{k=1}^K \nabla_\theta \log p_\theta(\boldsymbol{a}_t^{k-1} | \boldsymbol{a}_t^k, \boldsymbol{s}_t) \right]. \tag{11}$$

However, optimizing with Eq. (11) can be computationally intensive, as it requires generating new samples after each optimization step. To enhance sample efficiency and leverage historical sequences, we employ importance sampling, following the approach of proximal policy optimization (PPO) (Schulman et al., 2017), and derive the loss function for reward improvement as follows:

$$\mathcal{L}_{\text{Imp}}^{\mathcal{T}}(\theta) = \sum_t \mathbb{E}_{p_{\theta_{\text{old}}}(\boldsymbol{a}_t^{0:K} | \boldsymbol{s}_t)} \left[ \sum_{k=1}^K -r^{\mathcal{T}}(\boldsymbol{a}_t^0) \max \Big( \rho_k(\theta, \theta_{\text{old}}), \text{clip}\left(\rho_k(\theta, \theta_{\text{old}}), 1 + \epsilon, 1 - \epsilon\right) \Big) \right], \tag{12}$$

where $\rho_k(\theta, \theta_{\text{old}}) = \frac{p_\theta(\boldsymbol{a}_t^{k-1} | \boldsymbol{a}_t^k, \boldsymbol{s}_t)}{p_{\theta_{\text{old}}}(\boldsymbol{a}_t^{k-1} | \boldsymbol{a}_t^k, \boldsymbol{s}_t)}$ and $\epsilon$ is a hyperparameter. Then, we can train our model using $\mathcal{L}_{\text{Imp}}^{\mathcal{T}}(\theta)$ in an end-to-end manner, which is equivalent to maximizing the objective in Eq. (10).

**Regularization term.** However, fine-tuning the model solely depending on the reward is insufficient since the model may step too far, which can lead to performance collapse and instability during reward maximization. To address this problem, we introduce a Behavior-Clone (BC) regularization term during the fine-tuning process. Concretely, we aim to constrain our policy $\theta$ to closely match a target policy $\mu$, ensuring that $\theta$ does not deviate significantly from $\mu$ after policy updates. This constraint can be modeled using a negative log-likelihood (NLL) loss as:

$$\min_\theta \mathbb{E}_{\boldsymbol{a}_\mu^0 \sim p_\mu} \left[ -\log p_\theta(\boldsymbol{a}_\mu^0) \right]. \tag{13}$$

Following Ho et al. (2020), we can obtain a surrogate loss to optimize Eq. (13) as follows:

$$\mathcal{L}_{\text{BC}}(\theta) = \mathbb{E}_{k \sim [1,K], \boldsymbol{a}_\mu^k \sim p_\mu} \left[ \left\| \epsilon(\boldsymbol{a}_\mu^k, k) - \epsilon_\theta(\boldsymbol{a}_\mu^k, k) \right\|^2 \right], \tag{14}$$

where $\epsilon(\boldsymbol{a}_\mu^k, k)$ represents the ground-truth noise added to $\boldsymbol{a}_\mu^k$ at timestep $k$, which can be calculated as $\epsilon(\boldsymbol{a}_\mu^k, k) = \frac{\boldsymbol{a}_\mu^k - \sqrt{\bar{\alpha}_k} \cdot \boldsymbol{a}_\mu^0}{\sqrt{1 - \bar{\alpha}_k}}$.

**How to select the target policy?** Intuitively, an ideal target policy is the optimal policy that generates samples $x^*$ satisfying $\mathcal{C}(x^*) \geq \mathcal{C}(\boldsymbol{x})$ for all possible $\boldsymbol{x}$, where $\mathcal{C}(\boldsymbol{x})$ represents a measure of the performance or quality of the sample, such as the accumulated reward for action sequences. Since $\mu$ is unknown during fine-tuning, we approximate it by sampling action sequences $\boldsymbol{a}$ that satisfy $\mathcal{C}(\boldsymbol{a}) \approx \mathcal{C}(\boldsymbol{a}^*)$. In practice, we denote $\boldsymbol{a}^*$ as the best actions from recent play experience, such as those that yielded the top $n$ highest rewards or successfully completed the given task. We then sample $\boldsymbol{a}^k$ from these proficient action sequences obtained during online interaction, nearly equivalent to sampling from $\mu$ to regularize the fine-tuning process. We also remark that the BC regularizer

is not the only way to incorporate regularization into Eq. (12). For example, a Kullback–Leibler (KL) divergence between the fine-tuned and pre-trained models, or a diffusion pre-train loss can be employed to regularize the fine-tuning process, as shown in text-to-image and text-to-speech generation (Fan et al., 2024; Chen et al., 2024). However, we find these regularization may cause the pre-trained planner trap in sub-optimal regions, hindering performance improvement. We will further discuss them in experiments.

Combining Eq. (12) with Eq. (14), the loss function for reward fine-tuning in downstream tasks $\mathcal{T} \sim p(\mathcal{T})$ is expressed as follows:

$$\mathcal{L}^{\mathcal{T}}_{\text{fine-tuning}}(\theta) = \mathcal{L}^{\mathcal{T}}_{\text{Imp}}(\theta) + \lambda \mathcal{L}_{\text{BC}}(\theta), \tag{15}$$

where $\lambda$ is a weight coefficient. The overall process of pre-training and fine-tuning using SODP is summarized in Alg. 1 in the appendix. Since our goal is to generate complete trajectories rather than individual segments, we utilize a trajectory-level buffer (Zheng et al., 2022) for estimating the target policy $\mu$. Further, to ensure the accuracy of the approximation, we generate several proficient trajectories using the pre-trained model at the beginning of each iteration.

## 4 RELATED WORK

**Diffusion Models in RL.** Diffusion models are a leading class of generative models, achieving state-of-the-art performance across a variety of tasks, such as image generation (Ramesh et al., 2021), audio synthesis (Kong et al., 2020; Huang et al., 2023), and drug design (Schneuing et al., 2022; Guan et al., 2024). Recent studies have applied them in imitation learning to model human demonstrations and predict future actions (Li et al., 2024; Reuss et al., 2023). Other approaches have trained conditional diffusion models either as planners (Ajay et al., 2022; Brehmer et al., 2024) or policies (Hansen-Estruch et al., 2023; Kang et al., 2024). However, most of these efforts focus on single-task settings. While some recent works aim to extend diffusion models to multi-task scenarios, they often rely on additional conditions, such as prompts (He et al., 2024) or preference labels (Yu et al., 2024). These methods are limited by their dependence on expert data or explicit task knowledge. In contrast, our method learns broad action-sequence distributions from inferior data to enhance action priors, enabling effective generalization across a range of downstream tasks.

**Fine-tuning Diffusion Models.** Despite the impressive success of diffusion models, they often face challenges in aligning with specific downstream objectives, such as image aesthetics (Schuhmann et al., 2022), fairness (Shen et al., 2023), or human preference (Xu et al., 2024), primarily due to their training on unsupervised data. Some methods have been proposed to address this issue by directly fine-tuning models using downstream objectives (Prabhudesai et al., 2023; Clark et al., 2023), but they rely on differentiable reward models, which are impractical in RL since accurately modeling rewards with neural networks is quite costly (Kim et al., 2023). Other methods reformulate the denoising process as an MDP and apply policy gradients for fine-tuning (Black et al., 2023; Fan et al., 2024). However, they heavily depend on strong pre-trained models and have proven ineffective in our case. Our goal is to fine-tune a less powerful model that has been trained on inferior data.

Concurrent with our work, DPPO (Ren et al., 2024) also explores reward fine-tuning for refining RL diffusion planners. However, their approach focuses exclusively on single-task settings and allows access to expert demonstrations. In contrast, we train our model on multi-task data without the need for superior demonstrations. Additionally, we analyze the limitations of current regularization methods for versatile RL diffusion models and propose a new regularizer that improves the performance of sub-optimal pre-trained models.

## 5 EXPERIMENTS

In this section, we conduct experiments to evaluate our proposed method and address the following questions: (1) How does SODP's performance compare to current methods? (2) Can SODP scale to high-dimensional observation inputs? (3) How does SODP achieve higher rewards during online fine-tuning?

## 5.1 EXPERIMENTAL SETUP

We evaluate SODP in both state-based and image-based environments. We conduct experiments on the Meta-World benchmark (Yu et al., 2019) for both state-based and image-based tasks. We also perform image-based experiments on the Adroit benchmark (Rajeswaran et al., 2017).

**Meta-World.** The Meta-World benchmark comprises 50 distinct manipulation tasks, each requiring a Sawyer robot to interact with various objects. These tasks are designed to assess the robot's ability to handle different scenarios, such as grasping, pushing, pulling, and manipulating objects of varying shapes, sizes, and complexities. While the state space and reward functions differ across tasks, the action space remains consistent. Following recent studies (He et al., 2024; Hu et al., 2024), we extend all tasks to a random-goal setting, referred to as MT50-rand.

**Adroit.** The Adroit benchmark includes three dexterous manipulation tasks, requiring a 24-degree-of-freedom dexterous hand to solve complex challenges such as in-hand manipulation and tool use. The goals in this environment are also randomized. For Adroit, we use images as the observation to assess whether our method can scale to high-dimensional input.

**Datasets.** Following previous work (He et al., 2024), for Meta-World, we use a sub-optimal dataset comprising the first 50% of experiences (50M transitions) obtained from the replay buffer of a SAC (Haarnoja et al., 2018) agent during training. To verify the applicability of our method to tasks of varying difficulty levels, we divide the entire dataset into four subsets based on the task categories presented in Seo et al. (2023). For Adroit, we train a VRL3 (Wang et al., 2022a) agent for each task and use the initial 30% experiences (90K transitions) from the converged replay buffer. For Meta-World, all baselines and our pre-training stage are trained on the same dataset. For Adroit, the baselines are trained on expert demonstrations and ours is trained on sub-optimal transitions.

**Baselines.** For Meta-World, we compare our proposed SODP with the following baselines: (1) **MT-SAC.** Extended SAC with one-hot task ID as additional input. (2) **MTBC.** Extended BC to multi-task learning through network scaling and a task-ID-conditioned actor. (3) **MTIQL.** Extended IQL (Kostrikov et al., 2021) with multi-head critic networks and a task-ID-conditioned actor for multi-task policy learning. (4) **MTDQL.** Extended Diffusion-QL (Wang et al., 2022b) which is similar to MTIQL. (5) **MTDT.** Extended Decision Transformer (DT) (Chen et al., 2021a) to multitask settings by incorporating task ID encoding and state inputs for task-specific learning. (6) **Prompt-DT** (Xu et al., 2022). An extension of DT, which generates actions by utilizing trajectory prompts and reward-to-go signals. (7) **MTDIFF** (He et al., 2024). A diffusion-based approach that integrates Transformer architectures with prompt learning to facilitate generative planning in multitask offline environments. We extend it with a visual extractor in image-based Meta-World experiments. (8) **HarmoDT** (Hu et al., 2024). A DT-based approach that leverages parameter sharing to exploit task similarities while mitigating the adverse effects of conflicting gradients simultaneously. The results for these baselines are directly replicated from those reported in HarmoDT (Hu et al., 2024).

The action space for different tasks in Adroit is different and is incompatible with MTDIFF and HarmoDT. Therefore, we compare SODP with following baselines designed for complex environments: (1) **BCRNN** (Mandlekar et al., 2021). A variant of BC that employs a Recurrent Neural Network (RNN) as the policy network, predicting the sequence of actions based on the sequence of states as input. (2) **IBC** (Florence et al., 2022). Extended BC with energy-based models (EBM) to train implicit behavioral cloning policies. (3) **Diffusion Policy** (Chi et al., 2023). A diffusion-based approach that predicts future action sequences based on historical states. (4) **DP3** (Ze et al., 2024). A visual imitation learning algorithm that incorporates 3D visual representations into diffusion policies, using a point clouds encoder to process visual observations into visual features. The results for these baselines are directly replicated from those reported in DP3 (Ze et al., 2024).

## 5.2 RESULTS

We use the average success rate across all tasks as the evaluation metric and report the mean and standard deviation of success rates across three seeds. All baselines are trained on sub-optimal data. As shown in Table 1, our method achieves over a 60% success rate when learning from inferior data, outperforming all baseline methods. Compared to the existing state-of-the-art approach, our method demonstrates a 5.9% improvement. Notably, when compared to MTDIFF, the current leading method based on diffusion models, our approach shows a 24.4% improvement.

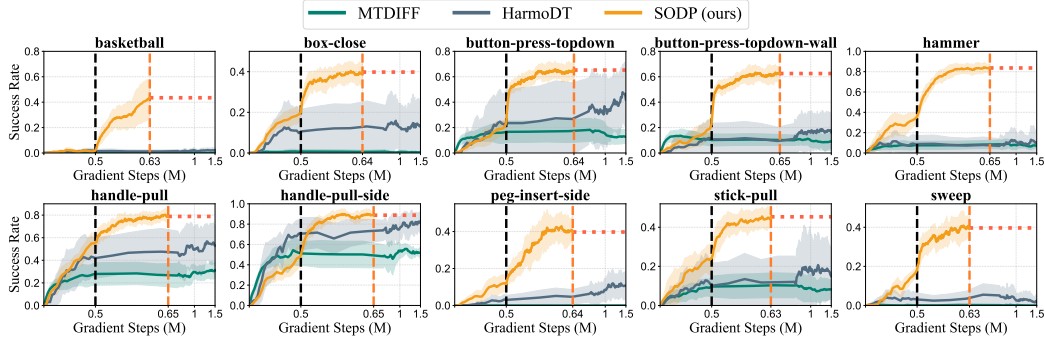

Figure 4: Learning efficiency. We sample 10 tasks and present the learning curves of SODP, MTD-IFF, and HarmoDT across five seeds. X-axis represents gradient steps. We pre-train the planner for $5e^5$ steps, followed by fine-tuning with a smaller number of steps. SODP rapidly converges to high success rates, whereas MTDIFF and HarmoDT struggle with some challenging tasks.

MTBC performs the worst, as imitation learning heavily depends on data quality, and directly cloning behaviors from sub-optimal data typically results in inferior performance. In contrast, our method models versatile action distributions from low-quality data and leverages them as priors to guide policy optimization in downstream tasks, leading to improved performance. We conduct additional experiments by augmenting original dataset with online trajectories and results can be found in Appendix C.1.

To further analyze the learning dynamics, we sample 10 tasks and present their learning curves of SODP alongside two leading baselines, MTDIFF and HarmoDT, across five seeds. As shown in Figure 4, SODP rapidly converges to high success rates, surpassing the other two baselines. The pre-training stage equips the planner with comprehensive action distribution priors and allows it to rapidly transfer and enhance these capabilities across a variety of downstream tasks. As a result, the pre-training stage significantly accelerates convergence, leading to

Table 1: Average success rate across 3 seeds on Meta-World 50 tasks with random goals (MT50-rand), using sub-optimal data. Each task is evaluated for 50 episodes.

| Method | Meta-World 50 Tasks |
|---|---|
| MTSAC | $42.67_{\pm 0.12}$ |
| MTBC | $34.53_{\pm 1.25}$ |
| MTIQL | $43.28_{\pm 0.90}$ |
| MTDQL | $17.33_{\pm 0.03}$ |
| MTDT | $42.33_{\pm 1.89}$ |
| Prompt-DT | $48.40_{\pm 0.16}$ |
| MTDIFF-P | $48.67_{\pm 1.32}$ |
| MTDIFF-P-ONEHOT | $48.94_{\pm 0.95}$ |
| HarmoDT-R | $53.80_{\pm 1.07}$ |
| HarmoDT-M | $57.20_{\pm 0.73}$ |
| HarmoDT-F | $57.20_{\pm 0.68}$ |
| **SODP (ours)** | $\mathbf{60.56}_{\pm 0.14}$ |

more efficient learning in the fine-tuning stage. The two baseline approaches struggle to address complex and challenging tasks such as *basketball* and *hammer*. In contrast, our method effectively guides the model to generate proficient actions, demonstrating the benefits of fine-tuning with policy gradient concerning return maximization. Moreover, while HarmoDT exhibits instability across different random seeds, our method demonstrates robustness against randomness.

**Does SODP generalize to high-dimensional observations?** We scale our method to image-based observations using the Adroit benchmark by employing a point-cloud encoder from DP3 (Ze et al., 2024) to process the 3D scene represented by point clouds. Specifically, we capture depth images directly from the environment and convert them into point clouds using Open3D (Zhou et al., 2018). These point clouds are then processed by the DP3 Encoder, which maps them into visual features. We then train our diffusion planner following the same procedure in Algorithm 1 except the input states are visual features. Following DP3 (Ze et al., 2024), We compute the average of the highest 5 evaluation success rates during training and report the mean and std across 3 seeds. As shown in Table 2, our method achieves an 8.2% improvement across all tasks. Since *hammer* is more challenging than

Table 2: Average success rate across 3 seeds on Adroit 3 tasks. IBC and BCRNN are extended by incorporating the DP3 point cloud encoder, resulting in IBD+3D and BCRNN+3D.

| Algorithm \ Task | Adroit | | | **Average** |
|---|---|---|---|---|
| | Hammer | Door | Pen | |
| BCRNN | $0_{\pm 0}$ | $0_{\pm 0}$ | $9_{\pm 3}$ | 3.0 |
| BCRNN+3D | $8_{\pm 14}$ | $0_{\pm 0}$ | $8_{\pm 1}$ | 5,3 |
| IBC | $0_{\pm 0}$ | $0_{\pm 0}$ | $9_{\pm 2}$ | 3.0 |
| IBC+3D | $0_{\pm 0}$ | $0_{\pm 0}$ | $10_{\pm 1}$ | 3.3 |
| Diffusion Policy | $48_{\pm 17}$ | $50_{\pm 5}$ | $25_{\pm 4}$ | 31.7 |
| Simple DP3 | $100_{\pm 0}$ | $58_{\pm 4}$ | $46_{\pm 5}$ | 68.0 |
| DP3 | $100_{\pm 0}$ | $62_{\pm 4}$ | $43_{\pm 6}$ | 68.3 |
| **SODP (ours)** | $67_{\pm 6}$ | $96_{\pm 1}$ | $59_{\pm 4}$ | **73.9** |

method achieves an 8.2% improvement across all tasks. Since *hammer* is more challenging than

*door*, our method may need more insightful priors from pre-training to achieve better performance. Experiments on image-based Meta-World can be found in Appendix C.5.

## 5.3 EFFECTIVENESS OF BC REGULARIZATION

To demonstrate the effectiveness of our BC regularization, we conduct an ablation study on fine-tuning same pre-trained model with our BC regularization and other variants. We consider following variants:

- **SODP w/o regularization.** This variant is similar to DDPO (Black et al., 2023) and DPPO (Ren et al., 2024), which fine-tunes the model directly using Eq. (12) without any regularization.

- **SODP_kl.** This variant is similar to DPOK (Fan et al., 2024), with the addition of a KL regularization term to constrain the divergence between the fine-tuned model and the pre-trained model.

- **SODP_pl.** This variant is similar to DLPO (Chen et al., 2024), incorporating the original diffusion pre-training loss (PL) into the fine-tuning objective to prevent the model from deviation.

The details of these variants are presented in Appendix E and more ablation studies on different fine-tuning methods can be found in Appendix C.2. Figure 5 demonstrates the effectiveness of our regularization in achieving a higher success rate. We observe that directly fine-tuning the model without any regularization results in the worst performance, with a decline in success rate, as the model may degrade the capabilities acquired from pre-training due to the lack of constraints. However, adding KL and PL is insufficient, as they cause oscillations near the pre-trained model. This aligns with the original intent of these regularizers, which is to prevent excessive deviation. This is reasonable for methods like DPOK and DLPO, which utilize pre-trained models such as Stable Diffusion (Rombach et al., 2022) and WaveGrad2 (Chen et al., 2021b). These models already exhibit strong generative capabilities without fine-tuning, and the goal is to make slight adjustments to align them with more fine-grained attributes, such as aesthetic scores and human preferences.

In contrast, our model is pre-trained on sub-optimal data and lacks the ability to solve complex tasks. We expect it to develop new skills for completing these tasks through fine-tuning. However, directly applying KL regularization to the pre-trained model leads to conservative policies that heavily rely on the existing capability, thereby confining the model to a sub-optimal region. While PL regularization allows some slight exploration, it is uncontrolled and random. Consequently, we observe that the KL regularization almost remains unchanged and the PL regularization slightly increases the performance in *basketball* but decreases in other tasks.

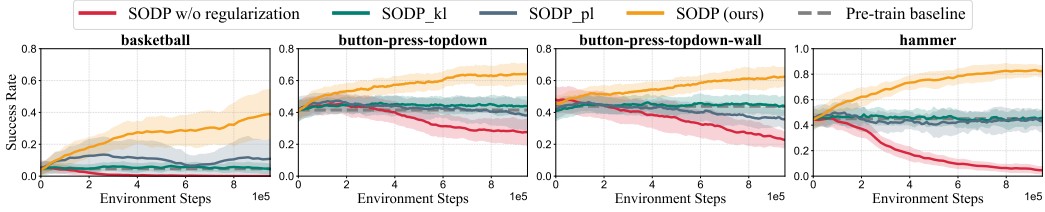

Figure 5: Learning efficiency for different regularization. X-axis represents environment steps. Performance declines significantly without any regularization. Both KL and PL regularization confine the model to sub-optimal regions. In contrast, our BC regularization effectively guides the model away from these sub-optimal areas, facilitating the attainment of optimal actions.

**Visualization.** We hypothesize that the effectiveness of our BC regularization lies in two aspects: (i) it ensures that our model can reuse the skills it has acquired, thereby preventing a decline in performance; (ii) It guides our model to effectively explore optimal regions due to the utilization of optimal $\mu$ as the target policy. To demonstrate this, we visualize trajectories of using the actions generated by our planner using t-SNE (Van der Maaten & Hinton, 2008). As shown in Figure 6, the trajectory distribution after fine-tuning with KL regularization closely resembles the original pre-training distribution, indicating that the model is reusing learned actions and lacks exploration into new regions. The exploration in PL is unstructured as it may lead to worse regions (e.g., the upper-left region in *basketball*). In contrast, our method demonstrates superior exploration capabilities to discover new, high-reward regions based on acquired knowledge (e.g., the lower-left region in

*basketball* and the bottom region in *plate-slide*). Meanwhile, the model can derive valuable insights from pre-trained knowledge by exploiting discovered high-reward actions (e.g. the central region in *plate-slide*) while discarding low-reward actions.

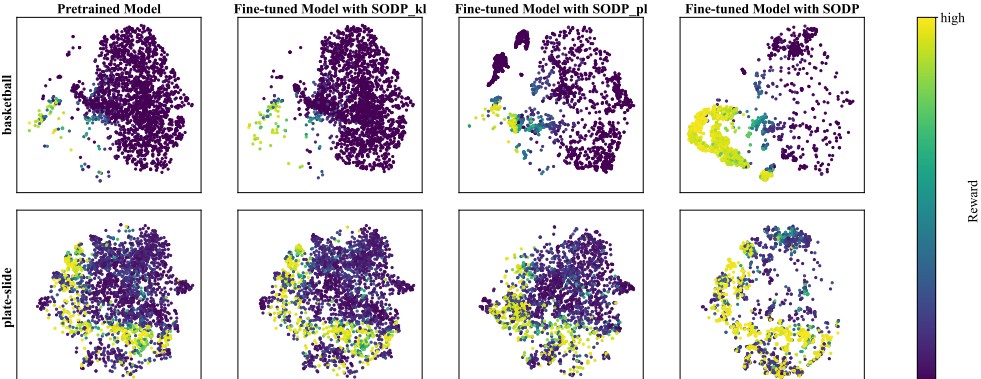

Figure 6: Visualization of trajectories using generated actions for different regularization. KL and PL regularization results in conservative policies with distributions closely resembling the original. Our BC regularization retains pre-trained knowledge while effectively discovering new actions that can lead to high rewards.

## 5.4 EFFECTIVENESS OF PRE-TRAINING

We investigate the impact of pre-training. We compare the performance of SODP with a version trained from scratch (**SODP_scratch**). For SODP_scratch, we use the same rollouts generated by the pre-trained model to approximate the target policy and initialize the replay buffer.

Figure 7 shows that fine-tuning the planner from scratch results in worse performance. Without pre-training, the planner lacks an action prior to guide its behavior, leading to stagnation as it struggles to move towards high reward regions. Additionally, it becomes unstable, as the limited useful knowledge is easily disrupted by a large number of ineffective trials.

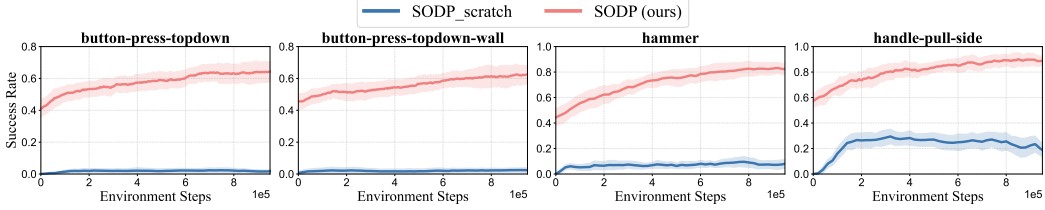

Figure 7: Effectiveness of pre-training. X-axis represents environment steps. Fine-tuning from scratch struggles to identify high-reward actions due to the lack of representation prior. In contrast, pre-training allows the planner to extract useful knowledge, guiding fine-tuning by refining the prior distribution towards more effective behaviors.

## 6 CONCLUSION

We propose SODP, a novel framework for training a versatile diffusion planner using sub-optimal data. By effectively combining pre-training and fine-tuning, we capture broad behavioral patterns drawn from large-scale multi-task transitions and then rapidly adapt them to achieve higher performance in specific downstream tasks. During fine-tuning, we introduce a BC regularization method, which preserves the pre-trained model's capabilities while guiding effective exploration. Experiments demonstrate that SODP achieves superior performance across a wide range of challenging manipulation tasks. In future work, we aim to develop embodied versatile agents that can effectively learn to solve real-world tasks using inferior data.

ETHICS STATEMENT

All procedures in this paper were conducted in accordance with the ICLR Code of Ethics (`https://iclr.cc/public/CodeOfEthics`).

REPRODUCIBILITY STATEMENT

We have provided all the implementation details necessary to reproduce our experiments in Appendix B, and the dataset used is the same as the one proposed by He et al. (2024).

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

# A  DERIVATIONS

## A.1  DERIVATION OF POLICY GRADIENT IN EQUATION (11)

Assume $p_\theta(\boldsymbol{a}_t^{0:K}|\boldsymbol{s}_t)r^\mathcal{T}(\boldsymbol{a}_t^0)$ and $\nabla_\theta p_\theta(\boldsymbol{a}_t^{0:K}|\boldsymbol{s}_t)r^\mathcal{T}(\boldsymbol{a}_t^0)$ are continuous (Fan et al., 2024), we have:

$$
\begin{aligned}
\nabla_\theta J^\mathcal{T}(\theta) &= \nabla_\theta \sum_t \mathbb{E}_{p_\theta(\boldsymbol{a}_t^0|\boldsymbol{s}_t)}\left[r^\mathcal{T}(\boldsymbol{a}_t^0)\right] \\
&= \sum_t \left[\nabla_\theta \int r^\mathcal{T}(\boldsymbol{a}_t^0)\cdot p_\theta(\boldsymbol{a}_t^0|\boldsymbol{s}_t)d\boldsymbol{a}_t^0\right] \\
&= \sum_t \left[\nabla_\theta \int r^\mathcal{T}(\boldsymbol{a}_t^0)\cdot \left(\int p_\theta(\boldsymbol{a}_t^{0:K}|\boldsymbol{s}_t)d\boldsymbol{a}_t^{1:K}\right)d\boldsymbol{a}_t^0\right] \\
&= \sum_t \left[\int r^\mathcal{T}(\boldsymbol{a}_t^0)\cdot \nabla_\theta \log p_\theta(\boldsymbol{a}_t^{0:K}|\boldsymbol{s}_t)\cdot p_\theta(\boldsymbol{a}_t^{0:K}|\boldsymbol{s}_t)\,d\boldsymbol{a}_t^{0:K}\right] \\
&= \sum_t \left[\int r^\mathcal{T}(\boldsymbol{a}_t^0)\cdot \nabla_\theta \log\left(p_K(\boldsymbol{a}_t^K|\boldsymbol{s}_t)\prod_{k=1}^K p_\theta(\boldsymbol{a}_t^{k-1}|\boldsymbol{a}_t^k,\boldsymbol{s}_t)\right)\cdot p_\theta(\boldsymbol{a}_t^{0:K}|\boldsymbol{s}_t)\,d\boldsymbol{a}_t^{0:K}\right] \\
&= \sum_t \mathbb{E}_{p_\theta(\boldsymbol{a}_t^{0:K}|\boldsymbol{s}_t)}\left[r^\mathcal{T}(\boldsymbol{a}_t^0)\sum_{k=1}^K \nabla_\theta \log p_\theta(\boldsymbol{a}_t^{k-1}|\boldsymbol{a}_t^k,\boldsymbol{s}_t)\right].
\end{aligned}
\tag{16}
$$

## A.2  DERIVATION OF LOSS FUNCTION IN EQUATION (12)

By using importance sampling approach, we can rewrite Eq. (16) as follows:

$$
\sum_t \mathbb{E}_{p_{\theta_{\text{old}}}(\boldsymbol{a}_t^{0:K}|\boldsymbol{s}_t)}\left[r^\mathcal{T}(\boldsymbol{a}_t^0)\sum_{k=1}^K \frac{p_\theta(\boldsymbol{a}_t^{k-1}|\boldsymbol{a}_t^k,\boldsymbol{s}_t)}{p_{\theta_{\text{old}}}(\boldsymbol{a}_t^{k-1}|\boldsymbol{a}_t^k,\boldsymbol{s}_t)}\nabla_\theta \log p_\theta(\boldsymbol{a}_t^{k-1}|\boldsymbol{a}_t^k,\boldsymbol{s}_t)\right]
\tag{17}
$$

Then, we can get a new objective function corresponding to Eq. (17) as:

$$
J_{\theta_{\text{old}}}^\mathcal{T}(\theta) = \max_\theta \sum_t \mathbb{E}_{p_{\theta_{\text{old}}}(\boldsymbol{a}_t^{0:K}|\boldsymbol{s}_t)}\left[r^\mathcal{T}(\boldsymbol{a}_t^0)\sum_{k=1}^K \frac{p_\theta(\boldsymbol{a}_t^{k-1}|\boldsymbol{a}_t^k,\boldsymbol{s}_t)}{p_{\theta_{\text{old}}}(\boldsymbol{a}_t^{k-1}|\boldsymbol{a}_t^k,\boldsymbol{s}_t)}\right]
\tag{18}
$$

Let $\rho_k(\theta,\theta_{\text{old}}) = \frac{p_\theta(\boldsymbol{a}_t^{k-1}|\boldsymbol{a}_t^k,\boldsymbol{s}_t)}{p_{\theta_{\text{old}}}(\boldsymbol{a}_t^{k-1}|\boldsymbol{a}_t^k,\boldsymbol{s}_t)}$ denote the probability ratio. Based on PPO (Schulman et al., 2017), we clip $\rho_k$ and use the minimum between the clipped and unclipped ratios to derive a lower bound of the original objective (18), which serves as our final objective function:

$$
J_{\text{clip}}^\mathcal{T}(\theta) = \max_\theta \sum_t \mathbb{E}_{p_{\theta_{\text{old}}}(\boldsymbol{a}_t^{0:K}|\boldsymbol{s}_t)}\left[r^\mathcal{T}(\boldsymbol{a}_t^0)\sum_{k=1}^K \min\Big(\rho_k(\theta,\theta_{\text{old}}),\text{clip}\left(\rho_k(\theta,\theta_{\text{old}}),1+\epsilon,1-\epsilon\right)\Big)\right]
\tag{19}
$$

To refine our pre-trained planner, we employ the negative of objective (19) as the loss function to facilitate reward maximization during fine-tuning.

### A.3 Derivation of loss function in Equation (14)

Directly computing and minimizing the NLL is difficult. However, we can derive an upper bound of Eq. (14) as follows:

$$\mathbb{E}_{\boldsymbol{a}_\mu^0 \sim p_\mu} \left[ -\log p_\theta(\boldsymbol{a}_\mu^0) \right] \leq \mathbb{E}_{\boldsymbol{a}_\mu^0 \sim p_\mu} \left[ \mathbb{E}_{q(\boldsymbol{a}_\mu^{1:K}|\boldsymbol{a}_\mu^0)} \left[ -\log \frac{p_\theta(\boldsymbol{a}_\mu^{0:K})}{q(\boldsymbol{a}_\mu^{1:K}|\boldsymbol{a}_\mu^0)} \right] \right]$$

$$= \mathbb{E}_{\boldsymbol{a}_\mu^0 \sim p_\mu} \left[ \mathbb{E}_{q(\boldsymbol{a}_\mu^{1:K}|\boldsymbol{a}_\mu^0)} \left[ -\log p(\boldsymbol{a}_\mu^K) - \sum_{k=1}^K \log \frac{p_\theta(\boldsymbol{a}_\mu^{k-1}|\boldsymbol{a}_\mu^k)}{q(\boldsymbol{a}_\mu^k|\boldsymbol{a}_\mu^{k-1})} \right] \right] \quad (20)$$

$$= \mathbb{E}_{\boldsymbol{a}_\mu^0 \sim p_\mu} \left[ \sum_{k=2}^K \mathbb{E}_{q(\boldsymbol{a}_\mu^k|\boldsymbol{a}_\mu^0)} D_{\mathrm{KL}}[q(\boldsymbol{a}_\mu^{k-1}|\boldsymbol{a}_\mu^k,\boldsymbol{a}_\mu^0)||p(\boldsymbol{a}_\mu^{k-1}|\boldsymbol{a}_\mu^k)] + \right.$$

$$\left. D_{\mathrm{KL}}(q(\boldsymbol{a}_\mu^K|\boldsymbol{a}_\mu^0)||p(\boldsymbol{a}_\mu^K)) - \mathbb{E}_{q(\boldsymbol{a}_\mu^1|\boldsymbol{a}_\mu^0)} \left[ \log p_\theta(\boldsymbol{a}_\mu^0|\boldsymbol{a}_\mu^1) \right] \right]$$

Following previous work (Ho et al., 2020), the optimization of the bound can be simplified as:

$$\arg\min_\theta \frac{1}{2\sigma_q^2(k)} \frac{(1-\alpha_k)^2}{(1-\bar{\alpha}_k)\alpha_k} \left\| \epsilon(\boldsymbol{a}_\mu^k,k) - \epsilon_\theta(\boldsymbol{a}_\mu^k,k) \right\|^2 \quad (21)$$

where:

$$\sigma_q^2(k) = \frac{(1-\alpha_k)(1-\bar{\alpha}_{k-1})}{1-\bar{\alpha}_k} \quad (22)$$

Here, $\epsilon_\theta(\boldsymbol{a}_\mu^k,k)$ is a noise model that learns to predict the source noise $\epsilon(\boldsymbol{a}_\mu^k,k)$ which determines $\boldsymbol{a}_\mu^k$ from $\boldsymbol{a}_\mu^0$.

## B The Details of SODP

### B.1 Diffusion Policy

We use diffusion policy (Chi et al., 2023) to generate future actions. For any given time step $t$, the model uses the most recent $T_o$ steps of states as input to generate the next $T_p$ action steps. Then, the first $T_a$ steps of these generated actions are executed in the environment without re-planning. In our experiments, we use $T_p = 12, T_o = 2, T_a = 8$ for Meta-World and $T_p = 4, T_o = 2, T_a = 3$ for Adroit.

We employ a CNN-based diffusion policy as our noise model, utilizing a U-net architecture that incorporates Feature-wise Linear Modulation (FiLM) (Perez et al., 2018) to condition on historical states. The implementation is based on the code from `https://github.com/CleanDiffuserTeam/CleanDiffuser`, and we use their default hyper-parameters. For Adroit, we use a simplified backbone provided by Simple DP3 (`https://github.com/YanjieZe/3D-Diffusion-Policy`), which removes some components in the U-net.

### B.2 Implementation Details

The pseudo-code of SODP is given in Alg. 1. We describe details of pre-training and fine-tuning as follows:

- For pretraining, we use cosine schedule for $\beta_k$ (Nichol & Dhariwal, 2021) and set diffusion steps $K = 100$. We pre-train the model for $5e^5$ steps in Meta-Wrold and $3e^3$ steps in Adroit.
- For fine-tuning, we use DDIM (Song et al., 2020) with 10 sampling steps and $\eta = 1$. We fine-tune each task for $1e^6$ steps in Meta-World and $3e^3$ steps in Adroit. Following DPOK (Fan et al., 2024), we perform $p_{\text{step}} \in \{10, 30\}$ gradient steps per episode. We set discount factor $\gamma = 1$ for all tasks.
- We set $N_{\text{init}} \in \{10, 20\}$ for approximating target distribution and $\lambda = 1.0$ as the BC weight coefficient.
- Batch size is set to 256 for both pre-training and fine-tuning.

- We use Adam optimizer (Kingma, 2014) with default parameters for both pre-training and fine-tuning. Learning rate is set to $1e^{-4}$ for pretraining and $1e^{-5}$ for fine-tuning with exponential decay.

---

**Algorithm 1:** SODP: Two-stage framework for learning from sub-optimal data

---

**Input:** diffsuion planner $\theta$, $N$ downstream tasks $\mathcal{T}_i$, multi-task sub-optimal data $D = \cup_{i=1}^{N} \mathcal{D}_{\mathcal{T}_i}$, target buffer $\mathcal{B}_{\text{target}}$, replay buffer $\mathcal{B}$, episode length L, pre-train $N_{\text{PT}}$ and fine-tune $N_{\text{FT}}$ steps

```
// pre-training model with sub-optimal data
```
**for** $t = 1, \ldots, N_{PT}$ **do**
    Sample $(s, a) \sim D$, diffusion time step $k \sim \text{Uniform}(\{1, \ldots, K\})$, noise $\epsilon \sim \mathcal{N}(0, \mathbf{I})$;
    Update $\theta$ using the loss function (7);
```
// fine-tuning model for downstream tasks
```
**for** $\mathcal{T}_i \in [\mathcal{T}_1, \ldots, \mathcal{T}_N]$ **do**
    **Initialization:** $\theta \leftarrow \theta_{\text{PT}}$; $\mathcal{B}, \mathcal{B}_{\text{target}} \leftarrow$ Rollout $N_{\text{init}}$ proficient trajectories using $\theta$;
    **for** $t = 1, \ldots, N_{FT}$ **do**
        **while** *not end of the episode* **do**
            Obtain samples $a_t^{0:K} \sim p_\theta(a_t^{0:K}|s_t)$;
            Execute the first $T_a$ steps and get reward $r(a_t^0)$;
            $\mathcal{B} \leftarrow \mathcal{B} \cup (s_t, a_t^{0:K}, r(a_t^0))$;
            $s_t \leftarrow s_{t+T_a}, t \leftarrow t + T_a$;
        ```
// approximate target policy μ
```
        **if** *proficient* **then**
            $\mathcal{B}_{\text{target}} \leftarrow \mathcal{B}_{\text{target}} \cup \{a_t^{0:K} | t \in \{0, T_a, \ldots, L\}\}$
        Compute $\mathcal{L}_{\text{Imp}}^{\mathcal{T}_i}$ using batches from $\mathcal{B}$ according to Eq. (12);
        Compute $\mathcal{L}_{\text{BC}}^{\mathcal{T}_i}$ using batches from $\mathcal{B}_{\text{target}}$ according to Eq. (14);
        Update $\theta$ using the loss function (15);

---

## C  EXTENDED RESULTS

In this section, we provide our full experimental results:

1. Baselines incorporating our online interaction trajectories as supplementary training data.

2. Ablation studies evaluating various fine-tuning strategies.

3. Analysis of the impact of pre-training dataset quality.

4. Generalizability to previously unseen tasks.

5. Evaluation on image-based Meta-World tasks across 10 environments.

### C.1  AUGMENTED TRAINING DATA FOR MTDIFF AND HARMODT

To isolate the influence of date quantity, we conducted fine-tuning for 100k steps per task using SODP, collecting online interaction samples during the fine-tuning process. These samples were then incorporated as a supplementary dataset alongside the original data, expanding the dataset size from 50M to 50M+100k×50. Subsequently, we trained both MTDIFF and HarmoDT on this augmented dataset to ensure consistent data usage across our method and the baseline methods. The experimental results are presented in Table 3, demonstrating that our method continues to outperform the baseline methods under this configuration. For MTDIFF, we employed the default parameters provided by the authors. However, a performance decline was observed on these new datasets, likely due to the increased presence of inferior data introduced during the online interaction phase.

Table 3: Average success rate using augmented sub-optimal data.

| Method | Meta-World 50 Tasks |
|---|---|
| MTDIFF-P | $27.06_{\pm 0.42}$ |
| HarmoDT-F | $57.37_{\pm 0.34}$ |
| **SODP (ours)** | $\mathbf{59.26}_{\pm 0.18}$ |

## C.2 ABLATION STUDIES EXAMINING OTHER FINE-TUNING APPROACHES

To demonstrate the effectiveness of our online fine-tuning approach, we compare it with two alternative fine-tuning methods: (i) **SODP_off**, which involves fine-tuning using high-quality offline data, and (ii) **SODP_off_scratch**, which performs direct training with high-quality data without pre-training. Specifically, we fine-tuned the pre-trained models for 100k steps across five tasks, collecting 200 successful episodes (equivalent to 100k steps) for each task. These datasets were then used to independently train five models in an offline setting, utilizing the same loss function as in Eq. (15) (**SODP_off**). Additionally, to investigate the impact of pre-training on offline fine-tuning, we trained the model directly without pre-training (**SODP_off_scratch**).

The experimental results, presented in Table 4, report the success rates averaged over three seeds. Without pre-training, the model lacks the necessary action priors to efficiently identify high-reward action distributions. Furthermore, directly fine-tuning with high-quality offline data proves insufficient, as static reward labels may fail to provide adequate guidance in dynamic environments, hindering the model's ability to facilitate efficient exploration.

Table 4: Average success rate for different fine-tuning approaches.

| Tasks | SODP_off | SODP_off scratch | SODP |
|---|---|---|---|
| button-press-topdown | $58.67_{\pm 0.03}$ | $40.67_{\pm 0.08}$ | $60.67_{\pm 0.03}$ |
| hammer | $71.33_{\pm 0.05}$ | $13.33_{\pm 0.06}$ | $73.33_{\pm 0.03}$ |
| handle-pull-side | $60.67_{\pm 0.03}$ | $42.67_{\pm 0.08}$ | $81.67_{\pm 0.07}$ |
| peg-insert-side | $25.33_{\pm 0.03}$ | $0.0_{\pm 0.0}$ | $32.67_{\pm 0.06}$ |
| handle-pull | $66.67_{\pm 0.03}$ | $31.33_{\pm 0.06}$ | $75.33_{\pm 0.04}$ |
| Average success rate | $56.53_{\pm 0.18}$ | $25.6_{\pm 0.18}$ | $64.73_{\pm 0.19}$ |

To highlight the importance of modeling the diffusion process as a MDP for reward fine-tuning, we consider an alternative approach that directly applies BC during fine-tuning, using only Eq. (14) as the loss function. As shown in Table 5, directly using BC results in poorer performance, as BC lacks reward labels to effectively guide exploration. While BC during the fine-tuning phase enables access to dynamic actions, it is limited to 'imitation' rather than 'evolution,' as the model is unable to differentiate between good and bad actions.

Table 5: Average success rate for directly BC during fine-tuning.

| Task | Directly BC | SODP |
|---|---|---|
| button-press-topdown | $51.3_{\pm 0.05}$ | $60.7_{\pm 0.03}$ |
| basketball | $21.3_{\pm 0.03}$ | $41.2_{\pm 0.16}$ |
| stick-pull | $26.7_{\pm 0.08}$ | $50.5_{\pm 0.04}$ |

## C.3 PRE-TRAINING USING NEAR-OPTIMAL DATA

To evaluate the impact of pre-training data quality on fine-tuning performance, we modified the near-optimal dataset provided by He et al. (2024) by retaining only the last 50% of the data. This modification ensured that the total number of transitions remained the same as the sub-optimal data used in the main paper, while significantly increasing the proportion of expert trajectories. We refer to this modified dataset as near-optimal data and pre-trained a model on the Meta-World 10 tasks. Subsequently, we followed the same fine-tuning procedure outlined in the main paper to fine-tune the model on each task. The experimental results are presented in Table 6. Incorporating more optimal data during the pre-training stage leads to better performance, as the model gains more priors about the optimal action distributions.

Table 6: Average success rate achieved after fine-tuning models pre-trained on different datasets.

| Tasks | Sub-optimal dataset | Near-optimal dataset |
|---|---|---|
| basketball | $52.67_{\pm 0.03}$ | $80.67_{\pm 0.03}$ |
| button-press | $88.00_{\pm 0.02}$ | $89.33_{\pm 0.03}$ |
| dial-turn | $80.67_{\pm 0.02}$ | $74.00_{\pm 0.04}$ |
| drawer-close | $100.00_{\pm 0.00}$ | $100.00_{\pm 0.00}$ |
| peg-insert-side | $62.67_{\pm 0.02}$ | $84.67_{\pm 0.02}$ |
| pick-place | $36.67_{\pm 0.03}$ | $59.33_{\pm 0.03}$ |
| push | $33.33_{\pm 0.03}$ | $50.67_{\pm 0.03}$ |
| reach | $68.67_{\pm 0.05}$ | $95.33_{\pm 0.01}$ |
| sweep-into | $60.67_{\pm 0.03}$ | $75.33_{\pm 0.01}$ |
| window-open | $69.33_{\pm 0.04}$ | $100.0_{\pm 0.00}$ |
| Average success rate | $65.27_{\pm 0.21}$ | $80.93_{\pm 0.16}$ |

### C.4 FINE-TUING ON UNSEEN TASKS

To evaluate the generalizability of SODP, we conduct experiments to fine-tune the model on tasks that were not included in the pre-training dataset. We pre-train a model on the MT-10 dataset (SODP_mt10) and fine-tune it on three tasks that are not present in the pre-training data. Additionally, to investigate the advantages of pre-training on a multi-task dataset versus a single-task dataset, we compare SODP_mt10 with a variant that is pre-trained solely on the *basketball* dataset (SODP_bas). As shown in Table 7, pre-training on multi-task data enhances generalizability to unseen tasks, as multi-task data provide a broader range of action distribution priors compared to single-task data.

Table 7: Average success rate achieved after fine-tuning on unseen tasks.

| Unseen tasks | SODP_mt10 | SODP_bas |
|---|---|---|
| drawer-open | $34.7_{\pm 0.06}$ | $0.0_{\pm 0.0}$ |
| plate-slide-side | $55.3_{\pm 0.33}$ | $0.0_{\pm 0.0}$ |
| handle-pull-side | $71.3_{\pm 0.13}$ | $0.0_{\pm 0.0}$ |

### C.5 EXPERIMENTS IN IMAGE-BASED META-WORLD

To further validate the scalability of SODP in handling high-dimensional observations, we conduct experiments on image-based Meta-World 10 tasks. Since no existing image-based sub-optimal dataset for Meta-World is available, we collect data for the 10 tasks by training separate SAC agents for each task, as done in He et al. (2024), and rendering the environments to obtain image data. We then follow the same procedure as in Adroit to convert the images into point clouds and use the DP3 encoder to extract visual features. For comparison, we consider the following baselines: DP3 and MT-DIFF_3D, an extended variant of MTDIFF that employs the same 3D visual encoder used in SODP. The experimental results are presented in Table 8, demonstrating the generalizability of our method to complex inputs.

Table 8: Average success rate of image-based MT-10 tasks.

| Methods | Success rate |
|---|---|
| DP3 | $32.6_{\pm 0.23}$ |
| MTDIFF_3D | $38.0_{\pm 0.82}$ |
| SODP | $47.5_{\pm 0.18}$ |

## D THE DETAILS OF BASELINES

We describe the details of baselines used for comparison in our experiments. For Meta-World, we consider following baselines:

- **MTSAC.** The one-hot encoded task ID is incorporated into the original SAC as an additional input.

- **MTBC.** The actor network is modeled using a 3-layer MLP with Mish activation. In training and inference, the scalar task ID is processed through a separate 3-layer MLP with Mish activation to produce a latent variable $z$. The input to the actor network is then formed by concatenating the original state with this latent variable $z$

- **MTIQL.** Similar to MTBC, the actor network incorporates the task ID through a task-aware embedding. A multi-head critic network is employed to estimate the $Q$-values for each task, with each head being parameterized by a 3-layer MLP using Mish activation.

- **MTDQL.** Similar to MTIQL, a multi-head critic network is utilized to predict the Q-value for each task, and the original diffusion actor is extended with an additional task ID input.

- **MTDT.** The task ID is embedded into a latent variable $z$ of size 12. This latent variable is then concatenated with the raw state to form the input tokens.

- **Prompt-DT.** Actions are generated based on trajectory prompts and the reward-to-go. A GPT-2 transformer model is utilized as the noise network.

- **MTDIFF.** Actions are generated by a GPT-based diffusion model that incorporates prompt learning to capture task knowledge. MTDIFF considers a variant: MTDIFF-ONEHOT, which replaces the prompt with a one-hot task ID. We borrow the official codes from `https://github.com/tinnerhrhe/MTDiff` and use their default hyper-parameters.

- **HarmoDT.** Incorporate trainable task-specific masks to address gradient conflict by identifying an optimal harmony subspace of parameters for each task. There are three variants of HarmoDT: HarmoDT-R, which keeps task masks unchanged; HarmoDT-F and HarmoDT-M utilize different methods to weight masks. We borrow the official codes from `https://github.com/charleshsc/HarmoDT` and use their default hyper-parameters.

For Adroit, we consider following baselines:

- **BCRNN.** A variant of BC that models the policy network as an RNN. The network is trained on temporal sequences of length $H$, denoted as $(s_t, a_t, ..., s_{t+H}, a_{t+H})$, to predict action sequences based on historical states.

- **IBC.** BC is represented as a conditional energy-based modeling problem, where implicit policies are trained to imitate expert demonstrations.

- **Diffusion Policy.** The generation of robot behaviors is formulated as a conditional denoising diffusion process, where the diffusion model predicts action sequences based on given observations as conditions.

- **DP3.** Diffusion Policy is extended by incorporating 3D visual representations. The 3D scenes from the environment are represented as point clouds, which are then cropped and downsampled to reduce redundant information. These processed point clouds are passed through an MLP to generate visual representations, which serve as conditions for the diffusion models.

For image-based Meta-World, we extended MTDIFF by integrating the same 3D visual encoder used in SODP to extract visual features from input point clouds.

## E  VARIANTS OF SODP

In Eq. (15), we introduce a BC regularization term to preserve the pre-trained knowledge and demonstrate its effectiveness compared to two existing regularization approaches presented in DPOK(Fan et al., 2024) and DLPO (Chen et al., 2024). Specifically, the regularization term $\mathcal{L}_{\text{KL}}$ used in DPOK is expressed as:

$$\mathcal{L}_{\text{KL}}(\theta) = \sum_{k=1}^{K} \text{KL}(p_\theta(x_{k-1}|x_k)||p_{\text{pre}}(x_{k-1}|x_k)). \qquad (23)$$

And the regularization term $\mathcal{L}_{\text{PL}}$ used in DLPO is expressed as:

$$\mathcal{L}_{\text{PL}}(\theta) = \mathbb{E}_{k\sim[1,K],p_\theta(x_{1:K})} \left[ \|\epsilon(x_k,k) - \epsilon_\theta(x_k,k)\|^2 \right]. \qquad (24)$$

These methods can be considered as different approaches to selecting the target policy in line with our analysis and can be seen as variants of Eq. (14), where DPOK selects $\mu = \theta_{\text{pre-train}}$ and DLPO selects $\mu = \theta$. The rationale behind their selection is based on the assumption that $\theta \approx \theta_{\text{pre-train}} \approx \theta^*$. This assumption is reasonable in text-to-image or text-to-speech tasks, as the pre-trained models they used are already strong and perform exceptionally well even without fine-tuning. However, this assumption does not apply to our pre-trained planner, as the model is trained on sub-optimal data. As a result, as shown in Section 5.3, these regularization methods may lead the pre-trained planner to be stuck in inferior regions, limiting its ability to improve performance.

## F    COMPARISON TO DPPO

We summarize some similarities and differences between our work and the concurrent work DPPO (Ren et al., 2024) as follows:

- Both DPPO and our approach formulate the diffusion policy denoising process as an MDP and use policy gradients to fine-tune the model for higher environment rewards.
- DPPO demonstrated that reward-based RL fine-tuning promotes effective exploration, which is consistent with our observations.
- While DPPO requires task-specific expert demonstrations for pre-training, our method pre-trains a foundation model capable of capturing useful behavior patterns from multi-task inferior data.
- We show that directly fine-tuning the pre-trained planner without any regularization, as done in DPPO, fails in the multi-task setting. We further analyze the limitations of current regularization methods and propose a novel BC regularization term. By employing our regularizer, the pre-trained model achieves higher success rates after fine-tuning.
- Unlike DPPO, we don't employ advantage estimator.

## G    SINGLE-TASK PERFORMANCE

We evaluate the performance for each task for 50 episodes. We report the average evaluated return of pre-trained and fine-tuned models in Table 9.

Table 9: Evaluated return of SODP pre-trained model and fine-tuned model for each task in MT50-rand. We report the mean and standard deviation for 50 episodes for each task.

| Tasks | Return of pre-trained model | Return of fine-tuned model |
|---|---|---|
| basketball-v2 | $133.5 \pm 100.7$ | $2347.1 \pm 580.8$ |
| bin-picking-v2 | $96.8 \pm 23.9$ | $602.7 \pm 72.8$ |
| button-press-topdown-v2 | $1405 \pm 20.3$ | $1679 \pm 25.9$ |
| button-press-v2 | $1397 \pm 15.6$ | $2452.7 \pm 89.3$ |
| button-press-wall-v2 | $1375 \pm 10.58$ | $2524.7 \pm 18.1$ |
| coffee-button-v2 | $293.2 \pm 12.1$ | $451.5 \pm 14.4$ |
| coffee-pull-v2 | $39.5 \pm 6.2$ | $117.9 \pm 23.3$ |
| coffee-push-v2 | $33.8 \pm 6.1$ | $273.3 \pm 36.6$ |
| dial-turn-v2 | $1217.7 \pm 239.3$ | $1557.3 \pm 226.7$ |
| disassemble-v2 | $237 \pm 117.2$ | $502 \pm 164.6$ |
| door-close-v2 | $3347.7 \pm 124.9$ | $4116.3 \pm 118.6$ |
| door-lock-v2 | $1042.3 \pm 94.9$ | $2491 \pm 79.5$ |
| door-open-v2 | $2036.3 \pm 79.3$ | $2460.3 \pm 57.7$ |
| door-unlock-v2 | $1335 \pm 46.9$ | $2257.7 \pm 323.0$ |
| hand-insert-v2 | $85.9 \pm 56.7$ | $449.5 \pm 54.9$ |
| drawer-close-v2 | $2468.3 \pm 167.2$ | $3953.7 \pm 214.3$ |
| drawer-open-v2 | $1656 \pm 45.6$ | $2489.7 \pm 188.4$ |
| faucet-open-v2 | $2728.7 \pm 424.1$ | $4094.7 \pm 290.3$ |
| faucet-close-v2 | $2156.7 \pm 113.6$ | $3772 \pm 70.1$ |
| handle-press-side-v2 | $1919.7 \pm 449.5$ | $3478.3 \pm 98.0$ |
| handle-press-v2 | $2216.3 \pm 182.0$ | $3415.7 \pm 221.6$ |
| handle-pull-side-v2 | $1351.7 \pm 119.0$ | $2665.7 \pm 243.9$ |
| handle-pull-v2 | $1510.7 \pm 111.6$ | $2734 \pm 64.3$ |
| lever-pull-v2 | $650.7 \pm 32.5$ | $1068.8 \pm 110.4$ |
| peg-insert-side-v2 | $300.3 \pm 122.2$ | $1969.7 \pm 237.6$ |
| pick-place-wall-v2 | $596.7 \pm 10.6$ | $1175.7 \pm 150.1$ |
| pick-out-of-hole-v2 | $38.5 \pm 6.3$ | $106.7 \pm 7.9$ |
| reach-v2 | $2664.3 \pm 77.5$ | $3083.7 \pm 149.7$ |
| push-back-v2 | $55.8 \pm 26.7$ | $350.9 \pm 30.3$ |
| push-v2 | $46.8 \pm 35.9$ | $148.9 \pm 43.9$ |
| pick-place-v2 | $3.9 \pm 0.2$ | $5.9 \pm 1.8$ |
| plate-slide-v2 | $1268.7 \pm 75.8$ | $2862 \pm 234.5$ |
| plate-slide-side-v2 | $826.4 \pm 54.3$ | $1929.7 \pm 104.5$ |
| plate-slide-back-v2 | $795.8 \pm 62.9$ | $1587.7 \pm 125.0$ |
| plate-slide-back-side-v2 | $626.7 \pm 36.9$ | $1541.3 \pm 78.5$ |
| soccer-v2 | $863.8 \pm 159.5$ | $1234.2 \pm 225.8$ |
| push-wall-v2 | $175.2 \pm 35.0$ | $471.7 \pm 73.9$ |
| shelf-place-v2 | $260.4 \pm 111.5$ | $785.1 \pm 81.5$ |
| sweep-into-v2 | $621.0 \pm 132.9$ | $1282.3 \pm 135.9$ |
| sweep-v2 | $442.3 \pm 83.0$ | $1081.7 \pm 58.0$ |
| window-open-v2 | $1342.3 \pm 60.1$ | $2474.3 \pm 266.4$ |
| window-close-v2 | $1087.7 \pm 78.9$ | $1816.7 \pm 166.3$ |
| assembly-v2 | $282.5 \pm 4.3$ | $446.1 \pm 26.9$ |
| button-press-topdown-wall-v2 | $1374 \pm 16.1$ | $1702.7 \pm 71.5$ |
| hammer-v2 | $1678.3 \pm 52.5$ | $1907.7 \pm 25.4$ |
| peg-unplug-side-v2 | $34.2 \pm 2.9$ | $52.9 \pm 4.6$ |
| reach-wall-v2 | $3373.7 \pm 41.7$ | $3839.7 \pm 69.3$ |
| stick-push-v2 | $412.9 \pm 95.8$ | $833.5 \pm 99.1$ |
| stick-pull-v2 | $1977 \pm 155.9$ | $3116.3 \pm 58.1$ |
| box-close-v2 | $692.3 \pm 22.8$ | $1300.1 \pm 53.2$ |

