# OpenReview forum: "Task-agnostic Pre-training and Task-guided Fine-tuning for Versatile Diffusion Planner"
_ICLR.cc/2025/Conference — Submitted to ICLR 2025_

### Official Review · Reviewer_V32c · 2024-10-30

**Soundness:** 2
**Presentation:** 3
**Contribution:** 2
**Rating:** 3
**Confidence:** 3

**Summary:**

The paper presents a novel framework, SODP (Sub-Optimal Diffusion Planner), designed for optimizing reinforcement learning (RL) models by integrating Proximal Policy Optimization (PPO) with a Behavior Cloning (BC) loss function. This approach leverages PPO to maximize task-specific returns during fine-tuning, while incorporating BC regularization to preserve competencies gained during the pre-training stage. The framework addresses the challenge of utilizing sub-optimal data, enabling effective policy adaptation across various downstream tasks.

The authors validate their methodology through extensive empirical evaluations in multi-task environments, demonstrating that SODP can efficiently adapt and learn from low-quality datasets. The experimental results show that the proposed framework consistently outperforms state-of-the-art methods, highlighting its ability to produce competitive outcomes despite the limitations of pre-training on sub-optimal data.

**Strengths:**

- Introduction of BC Surrogate Loss: The paper introduces a novel Behavior Cloning (BC) surrogate loss that effectively regularizes the fine-tuning process, ensuring stability and preserving pre-trained competencies.
- Effective Use of Sub-Optimal Data: SODP demonstrates an innovative approach for leveraging sub-optimal data, overcoming the need for high-quality labels while still achieving strong performance across tasks.
- Strong Empirical Results: The experimental results on Meta-World and Adroit benchmarks show clear advantages over state-of-the-art methods, highlighting the practical effectiveness of the proposed framework.

**Weaknesses:**

- Insufficient Explanation of Key Equations: The derivation of Equations (11) and (12) is not fully explained. Specifically, the summation of the log probabilities in the reverse diffusion process in Equation (11) lacks clarity, as it differs from the standard $p(a|s)$ formulation commonly used in reinforcement learning. This discrepancy requires further explanation to justify its relevance and correctness.
- Misapplication of Trust Region Loss: In Equation (12), the trust region loss is applied to the diffusion process rather than directly addressing the reinforcement learning process. This choice appears to overlook the key challenge of improving sample efficiency in reinforcement learning, which could undermine the effectiveness of the method in dealing with limited data.
- Over-reliance on BC Regularization: I suspect behavior cloning the better samples from the replay buffer starting from a diffusion model pre-trained on sub-optimal data is the key to the performance gain over baselines in the proposed method. Please consider an experiment with only the BC regularization term and no fine-tuning term. This is supported by the fact that SODP w/o regularization decrease in performance with online samples.

**Questions:**

- In equation (11), I assume $\sum_{k=1}^K \triangledown \log p_\theta (a_t^{k-1} | a^{k-1}_t, s_t)$ is modeling $ \triangledown p(a_t | s_t) $ in policy gradient for the RL MDP. The probability of a specific sample is known to be intractable in the diffusion process. Can you provide proof for calculating the gradient of probability of a specific sample in the diffusion process? Note, DDPO is applying the policy gradient to each step of the diffusion reverse process, where the probability of each reverse step is defined as an isomorphic gaussian distribution, whose probabilities are well-defined.
- In equation (12), is "generating new samples" (page 5, line 238) referring to samples in the RL MDP or the diffusion MDP? If it is in the RL MDP, how does applying the surrogate loss to the diffusion MDP help solving the RL MDP?

---

> ### Author Response · Authors · 2024-11-26
> **Response to Reviewer V32c**
>
> Dear Reviewer V32c,
>
> We sincerely appreciate your precious time and constructive comments. In the following, we would like to answer your concerns separately.
>
> **W1**: ... the summation of the log probabilities in the reverse diffusion process in Equation (11) lacks clarity, as it differs from the standard $p(a|s)$ formulation commonly used in reinforcement learning.
>
> **A1**: According to the formulation in Eq.(9), $\pi\_{\theta}(a|s)=p\_{\theta}(\symbfit{a}\_{t}\^{k-1}|\symbfit{a}\_{t}\^k,\symbfit{s}\_{t})$ which is consistent with the standard policy formulation. In other words, for the denoising MDP, we model the policy network $\pi$ as our pre-trained diffusion denoising network $p_{\theta}$. Similar to DDPO[1], when using DDIM sampler in our experiments, the denoising policy $\pi$ simplifies to well-defined isotropic Gaussian distributions through this formulation.
>
> **W2**: In Equation (12), the trust region loss is applied to the diffusion process rather than directly addressing the reinforcement learning process.
>
> **A2**: Since our diffusion functions as an action planner, the actions used for interacting with RL environments are sampled from the pre-trained diffusion model $p\_{\theta}$. Specifically, the action distribution is represented as $p(\symbfit{a}\_t|\symbfit{s}\_t)=p\_{\theta}(\symbfit{a}\_t^{0:K}|\symbfit{s}\_t)$. After applying the importance sampling trick, the old policy distribution becomes $p\_{\text{old}}(\symbfit{a}\_t|\symbfit{s}\_t)=p\_{\theta\_{\text{old}}}(\symbfit{a}\_t^{0:K}|\symbfit{s}\_t)$. Consequently, the trust-region loss for the RL MDP is equivalent to the loss for the diffusion MDP as defined in Eq. (12).
>
> **W3**: Over-reliance on BC Regularization ... consider an experiment with only the BC regularization term and no fine-tuning term ...
>
> **A3**: Thank you for your insightful suggestions. We conduct experiments using only Eq.(14) for fine-tuning, with the results presented below. Directly BC leads to poorer performance, as BC lacks reward labels to effectively guide exploration. Although BC during the fine-tuning phase can access dynamic actions, it is limited to 'imitation' rather than 'evolution,' since the model lacks the ability to differentiate between good and bad actions.
>
> | Task  |Directly BC|SODP|
> |----------|--------|--------|
> |button-press-topdown |51.3 ± 0.05| 60.7 ± 0.03|
> | basketball          |21.3 ± 0.03| 41.2 ± 0.16|
> | stick-pull          |26.7 ± 0.08| 50.5 ± 0.04|
>
> **Q1**: I assume $\sum_{k=1}^{K} \nabla\log p_{\theta}(a_{t}^{k-1}|a_{t}^k,s_{t})$ is modeling $\nabla p(a_t|s_t)$ in policy gradient for the RL MDP. The probability of a specific sample is known to be intractable in the diffusion process.
>
> **A4**: As discussed in A1, $p_{\theta}$ in Eq.(11) is our pre-trained diffusion denoising network. Therefore, $\sum_{k=1}^{K} \nabla\log p_{\theta}(a_{t}^{k-1}|a_{t}^k,s_{t})$ is modeling $\nabla \pi(a^k|s^k)$ in policy gradient for the diffusion MDP. Therefore, the calculation for the log-likelihoods is feasible since $\pi=p_{\theta}$ is a well-defined Gaussian distribution.
>
> **Q2**: In equation (12), is "generating new samples" referring to samples in the RL MDP or the diffusion MDP?
>
> **A5**: This refers to the samples in the diffusion MDP. Consequently, training with the loss function in Eq.(12) is equivalent to optimizing the diffusion model to focus on higher-reward actions.
>
> **References**
>
> [1] Black et al. "Training diffusion models with reinforcement learning." In ICLR 2024.

---

### Official Review · Reviewer_5i69 · 2024-11-02

**Soundness:** 2
**Presentation:** 2
**Contribution:** 2
**Rating:** 5
**Confidence:** 4

**Summary:**

This paper proposes a pre-training and fine-tuning method, SODP, for diffusion planner. The model is pre-trained on sub-optimal and multi-task data offline. Unlike reinforcement learning approaches, the pre-training stage does not require rewards. Additionally, the model is designed to operate without task descriptions or trajectory returns as conditions. The authors aim to show that generalizable capabilities can be acquired from multi-task sub-optimal data in pre-training, providing helpful knowledge for task-specific fine-tuning in an online reinforcement learning setting.

**Strengths:**

1. The approach of using sub-optimal, multi-task trajectories for pre-training to enhance downstream task performance after fine-tuning is interesting.
2. The paper offers a comprehensive discussion on various regularization techniques during fine-tuning, and the experiments show that the proposed BC-regularization method effectively balances reward maximization with preventing performance collapse.

**Weaknesses:**

1. Clarification is needed on whether both multi-task and sub-optimal data in pre-training equally contribute to improved fine-tuning performance for specific tasks.
    1. There is no experiment using only single-task sub-optimal data for pre-training. Although previous work [1] suggests that multi-task diffusion planners outperform single-task setups, the setting in this paper is different, involving both pre-training and fine-tuning. It remains unclear whether multi-task data specifically benefits downstream RL fine-tuning or if the model could already gain useful patterns from sub-optimal data by training on the same task as the target downstream task. If this is the case, unrelated tasks in pre-training may be redundant. Thus, experiments or analyses are needed to separate the contributions of multi-task data versus sub-optimal data, at least for a few representative tasks.
    2. The choice of using 50% and 30% of experiences collected from RL agents' replay buffers in Meta-World and Adroit needs to be explained. Additionally, results from varying the ratios of experience while keeping the total number of pre-training transitions fixed would be valuable. This analysis could clarify if there is a trade-off between fine-tuning performance and the quality of pre-training data (i.e. if using lower-quality data for pre-training negatively impacts fine-tuning). A comparison between fine-tuning results using expert data (100% experience) versus sub-optimal data with the exact transition count would be especially insightful.
2. Key baselines are missing, and the baselines used in state-based and image-based environments are inconsistent.
    1. Since online interactions with the environment are allowed in the fine-tuning stage, online RL methods should be included for a fair comparison. Currently, only BC and offline RL are used as baselines.
    2. Both BC and offline RL methods are compared in Meta-World, whereas offline RL methods are excluded in Adroit.
    3. Given that diffusion-based policies in BC, offline RL, and online RL [2-7] have shown strong performance compared to non-diffusion methods in prior work, including offline and online diffusion-based RL baselines would offer a more complete comparison for evaluating the fine-tuning approach with online RL for the diffusion model.
    4. Overall, the most critical baselines to prioritize, in both image and state-based environments, would be online RL (non-diffusion, such as PPO) and diffusion-based offline and online RL methods [2-7]. Including these baselines would allow for a fairer comparison by incorporating methods that offer strong modeling capabilities with diffusion and interaction with the environment.
3. The problem setting for sub-optimal data collection may lack precision, as in line 52, “sub-optimal data can be easily obtained in the real world.” Concrete examples or scenarios of how sub-optimal data might be collected in real-world settings would be helpful. Discussing potential challenges or limitations of these collection methods would provide readers with a clearer understanding of the motivation’s validity.
    *  Currently, the experimental setup collects sub-optimal data by training RL agents and treating part of the replay buffer as "inferior data." However, without rewards, identifying data that qualifies as sub-optimal could be challenging or even infeasible, especially since, as noted in line 153, “reward labels may be scarce or costly to obtain.” If rewards were readily accessible in the offline data or during interaction with the environment, then a diffusion planner could be directly trained using offline or online RL methods [2-7], and pre-training on reward-free sub-optimal data might no longer be necessary. This raises questions about the practicality of the motivation and problem setting proposed in the paper. It would be more meaningful if data with mixed or random levels of inferiority—without requiring a curated collection mechanism—could still enhance downstream fine-tuning during pre-training.
4. Several descriptions are either inconsistent or need further clarification:
    1. The claim in line 362, “The baselines used in Meta-World struggled to handle this high-dimensional data structure,” lacks supporting analysis or experimental evidence. Where in the paper is this conclusion demonstrated?
    2. Are all baselines for Meta-World evaluated in a multi-task setting, while those used in Adroit are in a single-task setting? Additionally, what does "Simple DP3" in Table 2 refer to? This term is not explained anywhere in the paper.
    3. There are inconsistencies in the experimental setup descriptions. In Sec. 5.1 (EXPERIMENTAL SETUP), line 344 states, “All baselines and our pre-training stage are trained on the same dataset” for Meta-World, while line 346 states, “All baselines are trained on expert demonstrations and our pre-training stage is trained on sub-optimal transitions” for Adroit. However, in Sec. 5.2 (RESULTS), line 376 says, “All baselines are trained on sub-optimal data.” Could you clarify which statement is correct? Do lines 344 and 346 mean that all baselines in Meta-World are trained on the same sub-optimal data used for SODP pre-training, while expert data are used in Adroit baselines? If so, what is the reason behind using different data sources for each environment?
    4. This could introduce unfairness if it is accurate that baselines are trained with sub-optimal data. For SODP fine-tuning, the model can interact with the environment to optimize actions guided by rewards, whereas the BC and offline-RL methods do not have this privilege. Additionally, while offline RL methods can leverage low-reward transitions, BC methods typically require expert-level data to imitate expert policies effectively. Sub-optimal data, especially when only 50% or 30% of the RL agents' replay buffer is used, likely does not approximate expert-level quality. It would be more convincing if offline-RL and BC baselines were trained with expert data or even allowed to interact with the environment, as in methods like GAIL [8].
5. Concerns regarding experimental results and setup:
    1. There is an imbalance in the number of tasks between the state-based and image-based environments: 50 tasks are used in Meta-World, while only 3 are used in Adroit. Why wasn’t image observation utilized in Meta-World to create an image-based environment there as well? Including more image-based tasks would provide a more comprehensive evaluation of SODP’s image-based performance and could clarify whether the performance drop observed in tasks like “Hammer” is an isolated case, as shown in Table 2. If SODP generally struggles with challenging image-based tasks, such as “Hammer,” this would indicate limited generalization ability.
    2. The claim in line 432—“training on inferior data is more difficult, and the baselines trained on expert data perform better in this case”—is not entirely convincing, as SODP has access to interact with the environment and obtain rewards in the fine-tuning stage, which should be advantageous even if the baselines are trained with expert data.
    3. Does the same trend appear in state-based environments, where SODP performs worse in challenging tasks, as observed in the image-based setting? Could you provide task-specific performance results from Table 1 instead of just the average success rate? This would allow for a more precise identification of any special cases.
    4. In Fig. 7, it appears unreasonable that SODP_scratch could not succeed despite being able to interact with the environment and optimize with rewards. Additional experiments with a diffusion planner trained from scratch using online RL methods [2] should be included to see if online RL methods also fail entirely. Training with online RL from scratch would be a fairer comparison for SODP_scratch.
    5. Line 511 states, “We use the same rollouts generated by the pre-trained model to approximate the target policy and initialize the replay buffer.” Could you clarify the methodology used here and discuss its validity for testing the performance of SODP_scratch?

**References**

[1] He et al. "Diffusion model is an effective planner and data synthesizer for multi-task reinforcement learning." In NeurIPS 2024.

[2] Yang et al. "Policy representation via diffusion probability model for reinforcement learning." arXiv 2023.

[3] Kang et al. "Efficient diffusion policies for offline reinforcement learning." In NeurIPS 2024.

[4] Chen et al. "Offline reinforcement learning via high-fidelity generative behavior modeling." In ICLR 2023.

[5] Hansen-Estruch et al. "Idql: Implicit q-learning as an actor-critic method with diffusion policies." arXiv 2023.

[6] Wang et al. "Diffusion policies as an expressive policy class for offline reinforcement learning." In ICLR 2023.

[7] Psenka et al. "Learning a diffusion model policy from rewards via q-score matching." In ICML 2024.

[8] Ho et al.. "Generative adversarial imitation learning." In NeurIPS 2016.

**Questions:**

1. Is the weight coefficient λ in Eq. 15 sensitive to changes? I could not find any implementation details or discussion regarding this hyperparameter. A sensitivity analysis for this hyperparameter should be provided to show how different values of λ affect the performance of their method.
2. In line 332, the statement “state space differs across tasks” could be clarified further. Does this mean that the dimensions of the state space differ across tasks or that the physical meanings of each index in the state space vary between tasks? If it refers to dimensional differences, how is this managed during the pre-training stage with multi-task data?
3. Could you clarify the meaning of line 968: “We show that directly fine-tuning the pre-trained planner without any regularization, as done in DPPO, fails in the multi-task setting”? As I understand it, fine-tuning task-specific rewards would shift to a single-task setting in the fine-tuning stage. Do you mean that the diffusion planner is actually fine-tuned on multiple tasks simultaneously?

---

> ### Author Response · Authors · 2024-11-26
> **Response to Reviewer 5i69 (Part 1/4)**
>
> Dear Reviewer 5i69,
>
> We sincerely appreciate your precious time and constructive comments. In the following, we would like to answer your concerns separately.
>
> **W1.1**: There is no experiment using only single-task sub-optimal data for pre-training.
>
> **A1.1**: Thank you for your insightful suggestion. There are two main reasons that we adpot multi-task data for per-training instaed of single-task:
> - We report the success rates achieved after fine-tuning models pre-trained on single-task sub-optimal data (SODP_single) as follows. For some challenging tasks, directly pre-training with single-task sub-optimal data can result in poorer performance and may fail to facilitate online fine-tuning (e.g. *basketball*).
>
> | Task  |SODP_single|SODP|
> |----------|--------|--------|
> | basketball   | 1.0 ± 0.01| 41.2 ± 0.16|
> | stick-pull   | 37.5 ± 0.09| 50.5 ± 0.04|
>
> - We conduct experiments to evaluate the performance of the pre-trained model on fine-tuning for unseen tasks. Specifically, the model is pre-trained on MT-10 data (SODP_mt10) as well as *basketball* data (SODP_bas). We then fine-tune the pre-trained model on 3 unseen tasks. The success rates achieved after fine-tuning is shown below:
>
> | Unseen task  |SODP_mt10|SODP_bas|
> |----------|--------|--------|
> | drawer-open        | 34.7 ± 0.06| 0.0 ± 0.0|
> | plate-slide-side   | 55.3 ± 0.33| 0.0 ± 0.0|
> | handle-pull-side   | 71.3 ± 0.13| 0.0 ± 0.0|
>
> In conclusion, pre-training on multi-task data not only facilitates downstream task learning but also improves generalizability to unseen tasks, as multi-task data provide a broader range of action distribution priors compared to single-task data.
>
> **W1.2**: The choice of using 50% and 30% of experiences ...  results from varying the ratios of experience while keeping the total number of pre-training transitions ...
>
> **A1.2**:
> Thank you for your insightful suggestions.
> - We utilize 30% experiences as it is sufficient to achieve strong performance on Adroit tasks. We also use 50% experiences for pre-training and the success rates of the fine-tuned models are shown below. We make a trade-off between fine-tuning performance and the quality of pre-training data
>
> | Methods  |Hammer|Door|Pen|Average
> |----------|--------|--------|--------|--------|
> | SODP_50%           | 78 ± 3| 100 ± 0| 65 ± 3| 81.2
> | SODP_30%           | 67 ± 6| 96 ± 1| 59 ± 4| 73.9
>
> - To maintain a consistent total number of pre-training transitions, we select the final 50% of the converged SAC data, instead of the initial 50% (used as the sub-optimal dataset), as a near-optimal pre-training dataset for the Meta-World 10 tasks. Following the procedure outlined in our paper, we report the success rates across three seeds as follows. Incorporating a higher proportion of optimal transitions can enhance performance.
>
> | Task  |Sub-optimal dataset|Near-optimal dataset|
> |----------|--------|--------|
> | basketball          |52.67 ± 0.03| 80.67 ± 0.03|
> | button-press        |88.00 ± 0.02| 89.33 ± 0.03|
> | dial-turn           |80.67 ± 0.02| 74.00 ± 0.04|
> | drawer-close        |100.00 ± 0.00| 100.00 ± 0.00|
> | peg-insert-side     |62.67 ± 0.02| 84.67 ± 0.02|
> | pick-place          |36.67 ± 0.03| 59.33 ± 0.03|
> | push                |33.33 ± 0.03| 50.67 ± 0.03|
> | reach               |68.67 ± 0.05| 95.33 ± 0.01|
> | sweep-into          |60.67 ± 0.03| 75.33 ± 0.01|
> | window-open         |69.33 ± 0.04| 100.0 ± 0.00|
> | Average success rate|65.27 ± 0.21| 80.93 ± 0.16|

---

> > ### Author Response · Authors · 2024-11-26
> > **Response to Reviewer 5i69 (Part 2/4)**
> >
> > **W2**: baselines ... online RL (non-diffusion, such as PPO) and diffusion-based offline and online RL methods.
> >
> > **A2**: Thank you for your insightful suggestions.
> > - For state-based Meta-World, we introduce MTSAC as a new online baseline and multi-task Diffusion-QL (MTDQL) as a new offline baseline. For MTSAC, we incorporate the one-hot encoded task ID as an additional input. For MTDQL, we employ a multi-head critic network to predict the Q-value for each task and extend the diffusion actor by adding a task ID input. For a fair comparision, we conducted fine-tuning for 100k steps per task in our method, collecting online interaction samples during the fine-tuning process. These samples were then incorporated as a supplementary dataset alongside the original data. Consequently, the resulting dataset expanded from 50M to 50M+100k×50. We subsequently trained MTDQL on this augmented dataset so that the data used for our method and MTDQL are same. We report the success rates across 3 seeds.
> >
> > | Methods  |MT-50 Success rate|
> > |----------|--------|
> > | MTSAC    | 42.67 ± 0.12|
> > | MTDQL    | 17.33 ± 0.03|
> > | SODP     | 59.26 ± 0.18|
> >
> > - For image-based environments, we extend MTDIFF with 3D visual encoder (MTDIFF_3D) as our new offline baselines. Due to the incompatibility of MTDIFF with the different action spaces of different tasks in Adroit, we applied this integration to the image-based Meta-World 10 task instead.
> >
> > | Methods  |Image-based Meta-World 10 success rate|
> > |----------|--------|
> > | DP3        | 32.6 ± 0.23|
> > | MTDIFF_3D  | 38.0 ± 0.82|
> > | SODP       | 47.5 ± 0.18|
> >
> >
> > **W3**: ... sub-optimal data by training RL agents and treating part of the replay buffer as "inferior data." ... and pre-training on reward-free sub-optimal data might no longer be necessary. ... the practicality of the motivation and problem setting ...
> >
> > **A3**: Thank you for your insightful questions. For pre-training, our method only requires that the dataset contains state-action transitions. Such data can be obtained through various means, including keyboard control in simulation environments or teleoperation for real-world robots. Using a replay buffer as transition dataset is one of the common collection methods. While we use this dataset for convenience and to facilitate comparison with existing baselines, our method can be applied to other data collection methods, provided that the final dataset contains state-action transitions.
> >
> > **W4.1**: “The baselines used in Meta-World struggled to handle this high-dimensional data structure" lacks supporting analysis or experimental evidence.
> >
> > **A4.1**: See A2
> >
> > **W4.2**: Are all baselines for Meta-World evaluated in a multi-task setting, while those used in Adroit are in a single-task setting? Additionally, what does "Simple DP3" in Table 2 refer to? This term is not explained anywhere in the paper.
> >
> > **A4.2**: Yes. In the Adroit environment, the action spaces of different tasks vary, making them incompatible with current multi-task baselines, which assume a consistent action space across tasks. Simple DP3 is a lightweight variant introduced in the DP3 paper which significantly reduces the number of parameters while retaining most of the learning capacity. To balance performance with computational resource consumption, we adopt Simple DP3 as the backbone model for our image-based experiments.
> >
> > **W4.3**: Do lines 344 and 346 mean that all baselines in Meta-World are trained on the same sub-optimal data used for SODP pre-training, while expert data are used in Adroit baselines? If so, what is the reason behind using different data sources for each environment?
> >
> > **A4.3**: We apologize for any cunfusion and you are correct. We have revised the descriptions to ensure consistency and clarity. Since Diffusion Policy and DP3 are both imitation learning approaches, we align with their original settings by using expert data. We train both Diffusion Policy and DP3 using our sub-optimal data, and report the corresponding success rates below. Additionally, we conduct further image-based experiments using the sub-optimal Meta-World 10 dataset, with additional details provided in A2.
> >
> > | Methods  |Hammer|Door|Pen|Average
> > |----------|--------|--------|--------|--------|
> > | Diffusion Policy   | 36 ± 11| 65 ± 3| 18 ± 4| 39.8
> > | DP3                | 47 ± 5 | 75 ± 3 | 36 ± 3| 52.7
> > | SODP               | 67 ± 6| 96 ± 1| 59 ± 4| 73.9

---

> ### Author Response · Authors · 2024-11-26
> **Response to Reviewer 5i69 (Part 3/4)**
>
> **W4.4**: ... unfairness ... baselines are trained with sub-optimal data ... offline-RL and BC baselines were trained with expert data or even allowed to interact with the environment ...
>
> **A4.4**:
> - To ensure a fair comparison, we conducted fine-tuning for 100k steps per task in our method, collecting online interaction samples during the fine-tuning process. These samples were then incorporated as a supplementary dataset alongside the original data. Consequently, the resulting dataset expanded from 50M to 50M+100k×50. We subsequently trained the baseline methods on this augmented dataset so that the data used for our method and baseline methods are same. The experimental results, presented below, demonstrate that our method still outperform the baseline methods under this configuration. For MTDIFF, we utilize the default parameters provided by the authors. However, we observe a performance decline on these new datasets, which may be attributed to the increased inclusion of sub-optimal data during our online interaction phase.
>
> | Methods  |MT-50 Success rate|
> |----------|--------|
> | MTDIFF   | 27.06 ± 0.42|
> | HarmoDT  | 57.37 ± 0.34|
> | SODP     | 59.26 ± 0.18|
>
> - Following [1], we apply MTBC to the entire replay buffer, which can be considered a near-optimal dataset due to the significant increase in the proportion of expert data. We adopt the MTBC results reported in [1] as follows. Our method achieves competitive performance compared to MTBC, which is trained on a larger amount of expert data.
>
> | Methods  |MT-50 Success rate|
> |----------|--------|
> | MTBC_near-optimal   | 60.39 ± 0.86|
> | SODP     | 60.56 ± 0.14|
>
> **W5.1**: Why wasn’t image observation utilized in Meta-World to create an image-based environment there as well?  Including more image-based tasks ...
>
> **A5.1**: Thank you for your insightful suggestions. Since no existing image-based sub-optimal dataset for Meta-World is available, we collect data for 10 tasks by training separate SAC agents for each task and rendering the environments to obtain image data. We then follow the same procedure as in Adroit to convert the images into point clouds and use the DP3 encoder to extract visual features. We use MTDIFF_3D, as described in A2, as our baseline, along with DP3 trained on our sub-optimal dataset. The experimental results are presented below.
>
> | Methods  |Image-based Meta-World 10 success rate|
> |----------|--------|
> | DP3        | 32.6 ± 0.23|
> | MTDIFF_3D  | 38.0 ± 0.82|
> | SODP       | 47.5 ± 0.18|
>
> **W5.2**: “training on inferior data is more difficult, and the baselines trained on expert data perform better in this case”—is not entirely convincing ...
>
> **A5.2**: Thank you for your insightful questions. We observed that incorporating additional expert data during pretraining improved performance on the Hammer task. This suggests that the poor performance may be attributed to an insufficient prior for the Hammer task, which could be an isolated case. We have revised our paper to clarify this and prevent any potential misunderstandings.
> | Methods  |Hammer|
> |----------|--------|
> | SODP_50%           | 78 ± 3|
> | SODP_30%           | 67 ± 6|
>
>
> **W5.3**: provide task-specific performance results from Table 1 instead of just the average success rate?
>
> **A5.3**: Thank you for your insightful questions. The varying difficulty levels of tasks impact their corresponding success rates. We provide several example tasks to compare the performance of our method with current baselines. While the success rates of all baselines decline when faced with more challenging tasks, our method consistently outperforms them. Moreover, our method can complete some tasks that baselines fail (e.g. *basketball*). The detailed performance for each task of our method is presented in Appendix G of our revised paper.
>
> | Tasks  |MTDIFF|HarmoDT|SODP|
> |----------|--------|--------|--------|
> | door-close    | 80.7 ± 0.12| 97.3 ± 0.05| 98.7 ± 0.02|
> | handle-pull   | 24.7 ± 0.09| 61.3 ± 0.12| 75.3 ± 0.04|
> | stick-pull    | 3.3 ± 0.02 | 32.0 ± 0.04| 50.5 ± 0.04|
> | coffee-push   | 0.0 ± 0.00 | 23.3 ± 0.01| 34.7 ± 0.03|
> | basketball    | 0.0 ± 0.00 | 4.0 ± 0.00 | 41.2 ± 0.16|

---

> > ### Author Response · Authors · 2024-11-26
> > **Response to Reviewer 5i69 (Part 4/4)**
> >
> > **W5.4**: ... SODP_scratch could not succeed despite being able to interact with the environment and optimize with rewards ... diffusion planner trained from scratch using online RL methods [2] ...
> >
> > **A5.4**: Thank you for your insightful questions. According to [3], the referenced paper [2] is categorized as a 'diffusion policy,' while our method is 'diffusion planner.' The key distinction lies in how the diffusion model is utilized: in diffusion policy, the model serves as an actor network and is trained using the traditional actor-critic paradigm. In contrast, diffusion planners do not rely on an additional critic network, and their training process aligns with the original noise-matching formulation. As a result, the performance of diffusion planners is highly dependent on the quality of the data, making it challenging to train a planner from scratch as we observed in SODP_scratch. Thus, online training of diffusion planners remains an open problem, as current diffusion planners are offline approaches, while online diffusion-based methods are diffusion policy.
> >
> > **W5.5**: Could you clarify the methodology used here and discuss its validity for testing the performance of SODP_scratch?
> >
> > **A5.5**: Thank you for your insightful questions. Our fine-tuning loss comprises two components: the reward-improvement loss and the BC loss. To approximate the target policy accurately, it is necessary to pre-collect some successful trajectories. With pretraining, this can be achieved using the pretrained model directly. However, without pretraining, the initial model is random and unable to generate any successful trajectories. Therefore, to isolate the influence of pretraining, it is essential to control for other variables, as the accuracy of the target policy approximation is crucial for calculating the BC loss correctly. To ensure consistency, we use the same initial approximation as in the standard SODP approach.
> >
> > **Q1**: Is the weight coefficient λ in Eq. 15 sensitive to changes?
> >
> > **A6**: Thank you for your insightful questions. We conducted experiments by varying λ and the experimental results are shown below. We found that a small λ may have a negative impact, while increasing λ does not produce a significant effect.
> >
> > | Tasks  |λ=0.1|λ=0.5|λ=3.0|λ=1.0 (default)|
> > |----------|--------|--------|--------|--------|
> > | hammer            | 61.3 ± 0.13| 64.7 ± 0.11| 72.7 ± 0.03 |73.33 ± 0.03|
> > | handle-pull-side  | 74.2 ± 0.10| 75.1 ± 0.10| 80.98 ± 0.05 |81.67 ± 0.07|
> >
> > **Q2** ...  that the dimensions of the state space differ across tasks or that the physical meanings of each index in the state space vary between tasks? ...
> >
> > **A7**: In Meta-World, the state dimensions remain consistent across tasks and the meaning of each index varies depending on the specific task.
> >
> > **Q3**: Could you clarify the meaning of line 968: “We show that directly fine-tuning the pre-trained planner without any regularization, as done in DPPO, fails in the multi-task setting”?
> >
> > **A8**: In DPPO, the model is fine-tuned using single-task pretraining, requiring a separate pretrained model for each task. In contrast, our approach utilizes a multi-task pretrained model, which is trained on a multi-task dataset. This allows us to fine-tune a single pretrained model across various tasks, eliminating the need for task-specific pretrained models.
> >
> > **References**
> >
> > [1] He et al. "Diffusion model is an effective planner and data synthesizer for multi-task reinforcement learning." In NeurIPS 2023.
> >
> > [2] Yang et al. "Policy representation via diffusion probability model for reinforcement learning." arXiv 2023.
> >
> > [3] Zhu et al. "Diffusion Models for Reinforcement Learning: A Survey." arXiv 2023.

---

> ### Comment · Reviewer_5i69 · 2024-11-30
>
> Thank you for the responses! They addressed some of my questions, and I have adjusted my score accordingly. However, a few answers remain unclear, and I have additional questions. If these points can be resolved, I would happily reconsider my score further.
>
> **Q1**:
>
> Could you provide more details on how multi-task pre-training is conducted on Adroit? You explicitly mentioned that "the action spaces of different tasks vary," but further clarification on how this is managed during pre-training would be helpful. Additionally, given the variation in action spaces across tasks in Adroit, how does multi-task pre-training transfer knowledge when the meaning of actions differs across tasks?
>
> **Q2**:
>
> I am unclear on how fine-tuning is performed across different environments. Please correct me if I misunderstand your evaluation methods. Specifically, are SODP fine-tuning experiments on state—and image-based Meta-World conducted in a single-task or multi-task setting? Suppose fine-tuning is done in a single-task setting. In that case, comparable baselines trained under single-task settings should also be included, as multi-task BC or offline RL methods do not always outperform single-task methods. This would make the comparisons with SODP fairer, especially when fine-tuned with single-task rewards.
>
> **Q3**:
>
> Why is DP [4], one of the state-of-the-art baselines for imitation learning, not included as a baseline in the Meta-World state-based setting? My understanding is that DP can be applied in both state- and image-based scenarios.
>
> **Q4**:
>
> > For some challenging tasks, directly pre-training with single-task sub-optimal data can result in poorer performance and may fail to facilitate online fine-tuning (e.g., basketball).
>
> Can you clarify whether most tasks perform better in multi-task pre-training than in single-task pre-training? Are the reported results specific to the state-based setting of Meta-World?
>
> **Q5**:
>
> > We apply MTBC to the entire replay buffer, which can be considered a near-optimal dataset due to the significant increase in the proportion of expert data
>
> > For Meta-World, all baselines and our pre-training stage are trained on the same dataset.
>
> Based on these statements, it is unsurprising that BC baselines underperform, as "50% of experiences" consist of both near-optimal and random actions, which is not how BC is typically used. BC baselines should be trained exclusively on expert data. SODP benefits significantly from its ability to collect data with rewards during the online RL fine-tuning stage, giving it an inherent advantage over BC. Even when BC is trained on pure expert data, SODP is expected to outperform it due to this advantage. However, given that MT-BC, trained on the entire replay buffer, performs comparably to SODP—despite the presence of suboptimal data—it raises an important question: If BC were trained solely on pure expert data, could it outperform SODP?
>
> **Q6**:
>
> Could you provide the return values of other baseline methods for a comprehensive comparison in Table 9? Including the success rates for SODP compared to all baselines would also be appreciated.

---

> > ### Comment · Reviewer_5i69 · 2024-11-30
> >
> > **Q7**:
> >
> > > According to [3], the referenced paper [2] is categorized as a 'diffusion policy,' while our method is a 'diffusion planner.' The key distinction lies in how the diffusion model is utilized.
> >
> > Could you further clarify the distinction between a diffusion policy and a diffusion planner? According to [5], "planning refers to generating trajectories, which can be either sequence of states or state-action pairs, to maximize the cumulative reward and selecting actions to track the trajectory," while "policy is typically a state-conditioned action distribution πθ (a|s)." From my understanding, the origin of diffusion planners can be traced to [6-7], where an entire trajectory (a sequence of state-action pairs) is generated, but only a single or a few actions are executed before replanning based on the observed state. Given this, I am uncertain if defining the backbone model in your paper as a planner rather than a policy is accurate. Specifically, the model appears more aligned with a diffusion policy [4], as its usage closely resembles the original DP. The primary distinction lies in your approach to executing a single-step action rather than utilizing the entire predicted action sequence as in the original method. If this interpretation is correct, it raises the question of why the diffusion policy described in [2] could not be used as an online RL baseline in your experiments. Could you provide further clarification on this distinction and the rationale behind your categorization?
> >
> >
> > **Q8**:
> >
> > > To maintain a consistent total number of pre-training transitions, we select the final 50% of the converged SAC data.
> >
> > Why does the "drawer-close" task perform worse after fine-tuning with more optimal data? Could you provide an explanation?
> >
> > **Q9**:
> >
> > > Such data can be obtained through various means, including keyboard control in simulation environments or teleoperation for real-world robots.
> >
> > From my perspective, the mentioned collection methods would generally be categorized as expert data rather than suboptimal or inferior data. One of the most exciting aspects of SODP is its ability to utilize data of any quality for pre-training, including highly suboptimal or even random-level data. Have you explored pre-training using only the first 10% or 20% of the data from the replay buffer in Meta-World? If so, could this approach still enhance fine-tuning performance or potentially outperform the baselines?
> >
> > **References**
> >
> > [4] Chi et al. "Diffusion Policy: Visuomotor Policy Learning via Action Diffusion." In RSS 2023.
> >
> > [5] Dong et al. "CleanDiffuser: An Easy-to-use Modularized Library for Diffusion Models in Decision Making." In NeurIPS 2024.
> >
> > [6] Janner et al. "Planning with Diffusion for Flexible Behavior Synthesis." In ICML 2022.
> >
> > [7] Ajay et al. "Is Conditional Generative Modeling All You Need for Decision-Making?" In ICLR 2023.

---

> ### Author Response · Authors · 2024-12-01
> **Response to Reviewer 5i69 (Part 1/3)**
>
> Dear Reviewer 5i69,
>
> Thank you so much for raising the score and for providing further feedback. Below, we would like to address your concerns step by step.
>
> **Q1**: Could you provide more details on how multi-task pre-training is conducted on Adroit?
>
> **A1**: As discussed in W4.2 and A4.2, we currently perform single-task pre-training for each task in Adroit. The purpose of this implementation is to verify the scalability of our method in complex, high-dimensional observation and action spaces. Adroit serves as a typical benchmark, controlling a more complex dexterous hand, as opposed to the simpler arm models in Meta-World.
>
> **Q2**: ... are SODP fine-tuning experiments on state—and image-based Meta-World conducted in a single-task or multi-task setting? ... comparable baselines trained under single-task settings should also be included ...
>
> **A2**: Thank you for your insightful questions. We fine-tune the pre-trained model in a single-task setting. We train a SAC in a single-task setting on MT-10 tasks and the results are shown below.
>
> | Task  |SODP|Single-task SAC|
> |----------|--------|--------|
> | basketball          |52.67 ± 0.03| 6.67 ± 0.03|
> | button-press        |88.00 ± 0.02| 100.00 ± 0.00|
> | dial-turn           |80.67 ± 0.02| 69.67 ± 0.08|
> | drawer-close        |100.00 ± 0.00| 100.00 ± 0.00|
> | peg-insert-side     |62.67 ± 0.02| 32.33 ± 0.07|
> | pick-place          |36.67 ± 0.03| 37.33 ± 0.09|
> | push                |33.33 ± 0.03| 22.67 ± 0.12|
> | reach               |68.67 ± 0.05| 85.67 ± 0.02|
> | sweep-into          |60.67 ± 0.03| 46.67 ± 0.11|
> | window-open         |69.33 ± 0.04| 100.0 ± 0.00|
> | Average success rate|65.27 ± 0.21| 60.01 ± 0.23|
>
> **Q3**: Why is DP, one of the state-of-the-art baselines for imitation learning, not included as a baseline in the Meta-World state-based setting?
>
> **A3** Thank you for your insightful questions. Directly using DP corresponds to our pre-training stage, which performs significantly worse than our method after fine-tuning. However, as noted in Q5, we previously applied DP only on a sub-optimal dataset. To address this, we conducted experiments using DP on pure expert data from the MT-10 tasks. The results are presented below.
>
> | Task  |SODP|DP_expert|
> |----------|--------|--------|
> | basketball          |52.67 ± 0.03| 41.33 ± 0.04|
> | button-press        |88.00 ± 0.02| 95.33 ± 0.02|
> | dial-turn           |80.67 ± 0.02| 74.33 ± 0.02|
> | drawer-close        |100.00 ± 0.00| 96.67 ± 0.01|
> | peg-insert-side     |62.67 ± 0.02| 55.33 ± 0.07|
> | pick-place          |36.67 ± 0.03| 27.67 ± 0.07|
> | push                |33.33 ± 0.03| 21.33 ± 0.06|
> | reach               |68.67 ± 0.05| 62.33 ± 0.09|
> | sweep-into          |60.67 ± 0.03| 54.67 ± 0.12|
> | window-open         |69.33 ± 0.04| 88.67 ± 0.06|
> | Average success rate|65.27 ± 0.21| 61.76 ± 0.26|
>
> **Q4**: Can you clarify whether most tasks perform better in multi-task pre-training than in single-task pre-training? Are the reported results specific to the state-based setting of Meta-World?
>
> **A4**: Yes, we observe that pre-training on multiple tasks facilitates fine-tuning more effectively than single-task pre-training, resulting in higher performance on average. We also find the same trend in image-based Meta-World environments.
>
> **Q5**: ... SODP benefits significantly from its ability to collect data with rewards during the online RL fine-tuning stage, giving it an inherent advantage over BC ... If BC were trained solely on pure expert data, could it outperform SODP?
>
> **A5**: Thank you for your insightful questions. We train an MTBC model using pure expert data from the MT-10 tasks, and the results are presented below.
>
> | Task  |SODP|MTBC_expert|
> |----------|--------|--------|
> | basketball          |52.67 ± 0.03| 6.67 ± 0.03|
> | button-press        |88.00 ± 0.02| 100.00 ± 0.00|
> | dial-turn           |80.67 ± 0.02| 65.33 ± 0.11|
> | drawer-close        |100.00 ± 0.00| 100.00 ± 0.00|
> | peg-insert-side     |62.67 ± 0.02| 64.00 ± 0.1|
> | pick-place          |36.67 ± 0.03| 45.33 ± 0.08|
> | push                |33.33 ± 0.03| 14.00 ± 0.09|
> | reach               |68.67 ± 0.05| 74.67 ± 0.12|
> | sweep-into          |60.67 ± 0.03| 52.67 ± 0.15|
> | window-open         |69.33 ± 0.04| 92.67 ± 0.09|
> | Average success rate|65.27 ± 0.21| 61.53 ± 0.32|

---

> > ### Author Response · Authors · 2024-12-01
> > **Response to Reviewer 5i69 (Part 2/3)**
> >
> > **Q6**: Could you provide the return values of other baseline methods for a comprehensive comparison in Table 9? Including the success rates for SODP compared to all baselines would also be appreciated.
> >
> > **A6**: Thank you for your insightful suggestions. We can provide the returns and success rates for MTDIFF and HarmoDT. For brevity, we list the results for 10 tasks below.
> >
> > | Task  |SODP_ret|SODP_suc|MTDIFF_ret|MTDIFF_suc|HarmoDT_ret|HarmoDT_suc|
> > |----------|--------|--------|--------|--------|--------|--------|
> > | basketball      |2476.9±1144.10|41.2±0.16|32.2±27.93|0.0±0.00|111.8±48.33|4.0±0.00
> > | dial-turn       |1624.6±122.61|59.7±0.12|1296.5±26.53|15.1±0.07|1217.2±105.61|4.7±0.08
> > | door-close      |3156.9±120.01|96.3±0.02|3357.5±735.86|78.0±0.17|3081.6±78.57|94.7±0.06
> > | drawer-close    |3953.7±231.56|100.0±0.00|4812.5±11.46|100.0±0.00|4387.2±331.04|97.3±0.05
> > | hammer          |1619.5±118.09|73.3±0.03|1414.6±175.28|9.3±0.09|2623.4±216.22|47.6±0.06
> > | handle-pull     |2719.9±550.89|75.3±0.04|505.3±55.19|19.3±0.01|2148.33±838.57|61.3±0.12
> > | peg-insert-side |1798.2±180.31|32.7±0.06|15.6±7.81|0.0±0.00|1907.2±744.32|48.0±0.14
> > | stick-pull      |3223.1±705.70|50.5±0.04|285.9±217.88|2.0±0.00|1921.9±1241.80|12.0±0.04
> > | sweep-into      |1874.7±159.81|66.3±0.07|398.1±405.5|8.7±0.11|1806.8±531.07|54.7±0.15
> > | window-close    |2002.6±142.71|98.3±0.01|1041.1±576.1|19.3±0.15|1979.7±255.2|100.0±0.00
> >
> > **Q7**: Could you provide further clarification on this distinction and the rationale behind your categorization?
> >
> > **A7**: Thank you for your insightful questions. First, we would like to clarify that our method also executes the first $T\_a$ steps without replanning and computes the accumulated rewards $r(a\_{t:t+T\_a})$ as $r(\symbfit{a}\_t^0)$ in Eq. (12).
> > The concept of the "planning" problem can be traced back to [6], where the diffusion planner essentially solves the "trajectory optimization" problem, which is defined as: $\symbfit{a}^*\_{0:T}=\arg\max\_{\symbfit{a}\_{0:T}} \sum\_{t=0}^Tr(s\_t,a\_t)$. The objective function $\max\_{\theta} \mathbb{E}\_{\tau \sim D} \left( \log p\_{\theta}(\symbfit{x}\_0 | \symbfit{y}(\tau)) \right)$ in [7] is also a variant of this formula, depending on the return condition. Thus, we view the "planner" as a diffusion generation process that explicitly considers the expected return throughout the denoising process. In contrast, the "policy" is solely aimed at replacing traditional RL policies to model $\pi(a | s)$, and there is no explicit return condition within it. Therefore, we do not expect a policy alone to achieve higher rewards. To facilitate return maximization, current methods often propose incorporating action improvement from traditional RL by using additional critic networks as external guidance. In conclusion, a "planner" explicitly refines the denoising process by incorporating the return signal, whereas a "policy" alone cannot achieve return maximization. According to the objective function in Eq. (10), our method is more aligned with the concept of a "planner" because it can refine the denoising process by incorporating the return signal.

---

> ### Author Response · Authors · 2024-12-01
> **Response to Reviewer 5i69 (Part 3/3)**
>
> **Q8**: Why does the "drawer-close" task perform worse after fine-tuning with more optimal data? Could you provide an explanation?
>
> **A8**: According to the results shown in A1.2, the success rate of "drawer-close" are the same (100.0). Did you mean the "dial-turn"? If so, we suspect that this may result from "overfitting" to the reward function. Specifically, at the beginning of the fine-tuning stage, the return and success rate of the sub-optimal pre-trained model are 0.68 and 1056, respectively, while those of the near-optimal pre-trained model are 0.38 and 2477. Since the replay buffer is collected by a SAC agent whose goal is to maximize return, models pre-trained on the near-optimal dataset achieve higher returns. However, higher reward does not necessarily equate to a higher success rate, even though there is a positive correlation between the two. Furthermore, we increased the number of fine-tuning steps and found that fine-tuning on the near-optimal dataset for the "dial-turn" task requires more steps to achieve the same performance as sub-optimal pre-trained model.
>
> **Q9**: Have you explored pre-training using only the first 10% or 20% of the data from the replay buffer in Meta-World? If so, could this approach still enhance fine-tuning performance or potentially outperform the baselines?
>
> **A9**: Thank you for your insightful questions. We conduct experiments using the first 10% and 20% of the data from the replay buffer across 5 Meta-World tasks. The success rates after pre-training and fine-tuning are presented below. Pre-training on a smaller percentage of data can still enhance downstream fine-tuning. As the amount of pre-training data increases, the fine-tuning success rate also improves.
>
> | Tasks  |10% pre-training|10% fine-tuning|20% pre-training|20% fine-tuning|
> |----------|--------|--------|--------|--------|
> | button-press        |12.67 ± 0.04| 28.00 ± 0.03|37.67 ± 0.03|60.67 ± 0.03|
> | dial-turn           |45.33 ± 0.02| 60.67 ± 0.04|48.33 ± 0.03|73.33 ± 0.04|
> | drawer-close        |52.67 ± 0.03| 84.33 ± 0.03|81.33 ± 0.04|97.33 ± 0.02|
> | sweep-into          |3.67 ± 0.02| 12.33 ± 0.05|3.67 ± 0.02|12.67 ± 0.03|
> | window-open         |5.33 ± 0.02| 20.67 ± 0.03|6.33 ± 0.01|22.33 ± 0.05|
> | Average success rate|23.93 ± 0.15| 41.2 ± 0.23| 35.47 ± 0.17 |53.27 ± 0.22

---

> > ### Comment · Reviewer_5i69 · 2024-12-03
> >
> > Thank you for your detailed response. After careful consideration, I have decided to maintain my current rating. From my perspective, the paper is not yet fully developed and requires further refinement. The motivation behind the work is valid and compelling. However, while the updated version of the paper in the rebuttal addresses some aspects, several issues remain unaddressed in the paper. I outline my primary concerns below:
> >
> > 1. Experimental Settings are Confusing
> >     1. The Adroit environment does not support multi-task pre-training, limiting experiments to single-task sub-optimal data pre-training and single-task online RL fine-tuning. This weakens the proposed claim that multi-task sub-optimal pre-training benefits image-based environments. While including experiments in Adroit is valid, their current description is unclear, leading to confusion among multiple reviewers. Additionally, given Adroit’s natural limitations, I recommend adopting other environments with unified action spaces better suited for multi-task, image-based settings, such as ManiSkill or LIBREO.
> >     2. Since SODP fine-tuning involves single-task online RL fine-tuning, single-task baselines should be included. This is important because multi-task RL and BC settings may not consistently outperform single-task setups.
> >     3. Baseline comparisons for BC should use pure expert data. SODP should demonstrate superior performance in these settings, particularly where access to online RL fine-tuning is allowed, unlike in BC.
> >     4. For image-based tasks, the baselines should include online RL, offline RL, and BC methods for consistency and comprehensive evaluation.
> > 2. Deeper Analysis of SODP's Assumptions and Effectiveness
> >     1. The additional results show that SODP underperforms compared to BC with pure expert data and online RL baselines in some tasks. A thorough analysis of why SODP works in certain tasks but not others is crucial for completeness.
> >     2. Under the settings using the first 10% or 20% of replay buffer data, results show performance degradation compared to the baselines. While this is expected, it raises concerns about the core motivation. The authors previously stated that "keyboard control in simulation environments or teleoperation" would be treated as expert data. However, if data collected this way qualifies as expert data, it contradicts the motivation to leverage sub-optimal data for pre-training. Furthermore, if replay buffer data at 10% or 20% cannot outperform the baselines, it casts doubt on the general viability of sub-optimal data, irrespective of its quality, to consistently benefit online RL fine-tuning after pre-training. Additional discussion is needed to clarify the scenarios where SODP will likely succeed.

---

### Official Review · Reviewer_kFg1 · 2024-11-03

**Soundness:** 2
**Presentation:** 3
**Contribution:** 2
**Rating:** 6
**Confidence:** 3

**Summary:**

This paper introduces SODP, a two-stage framework for training versatile diffusion planners for robotic manipulation tasks. The framework first pre-trains a diffusion model using large-scale sub-optimal data from multiple tasks without reward labels, then fine-tunes it for specific downstream tasks using policy gradient optimization with a novel BC regularization.

**Strengths:**

Pros:

- The paper is well-written with clear organization and easy to follow.

- The fine-tuning stage presents a systematic design, particularly in its BC regularization mechanism that effectively balances between preserving pre-trained knowledge and exploring new high-reward behaviors, supported by comprehensive ablation studies comparing different regularization strategies.

**Weaknesses:**

Cons:
1. While the paper claims using sub-optimal data for pre-training as an innovation, there appears to be no specialized design or methodological advancement in handling such data. The pre-training stage follows standard diffusion model training procedures, identical to those used with expert demonstrations.
2. A significant concern arises regarding the fairness of experimental comparisons. First, the learning curves in Figure 4 only display fine-tuning steps while omitting SODP's substantial pre-training phase (5e5 steps), potentially understating the total computational requirements. Second, the paper fails to clarify whether baseline methods undergo pre-training or have access to online data collection during training. This is particularly important as SODP benefits from continuous online data collection to build its target policy, which may give it an unfair advantage if baselines are limited to offline data.
3. A fundamental limitation is that the paper fails to isolate whether the performance gains stem from sub-optimal data pre-training or from the online data collection during fine-tuning. Critical control experiments are missing, particularly (1) fine-tuning with high-quality offline data and (2) direct training with high-quality data without pre-training. Without these comparisons, it remains unclear if the proposed two-stage framework and the use of sub-optimal data are truly necessary, as the improvements might simply result from the continuous collection of high-quality samples during online fine-tuning.
4. A direct empirical comparison between optimal and sub-optimal pre-training data is missing. Specifically, the paper should compare the current approach (using the first 50% of SAC training data) with a variant using high-reward trajectories (e.g., the last 30% of converged SAC data) for pre-training. This would help validate whether sub-optimal data is truly sufficient or if optimal data would lead to substantially better performance.
5. Why not directly BC in finetune phase. Since the method already collects high-reward trajectories during online interaction for BC regularization, why not directly fine-tune the pre-trained model using imitation learning on these trajectories? This simpler approach might achieve similar performance without the complexity of MDP formulation and policy gradient optimization. Without this comparison, it's unclear whether the proposed fine-tuning framework provides substantial benefits over straightforward behavior cloning of high-reward samples.

**Questions:**

See weakness.
I believe that more experiments are needed to illustrate the soundness and Contribution of the paper's SETTINGS and Designs.

---

> ### Author Response · Authors · 2024-11-26
> **Response to Reviewer kFg1 (Part 1/2)**
>
> Dear Reviewer kFg1,
>
> We sincerely appreciate your precious time and constructive comments. In the following, we would like to answer your concerns separately.
>
> **W1**: ... no specialized design or methodological advancement in handling such data. The pre-training stage follows standard diffusion model training procedures ...
>
> **A1**: Thank you for your insightful questions. In pre-training, our objective is to learn useful priors from extensive transition data to facilitate downstream task learning. Given the powerful multi-modal modeling capabilities of diffusion models, we found that a standard diffusion model is sufficient for effectively learning from our multi-task dataset. Thus, we avoided incorporating additional complexities, ensuring simplicity, scalability, and ease of use.
>
> **W2**: ... the fairness of experimental ... (1) the learning curves in Figure 4 ... (2) whether baseline methods undergo pre-training or have access to online data collection during training.
>
> **A2**:
> - The pre-training phase is also depicted in Figure 4. The learning curve corresponding to pre-training is shown prior to the vertical black dashed line, while the segment between the black dashed line and the orange dashed line represents the fine-tuning phase.
> - To ensure a fair comparison, we conducted fine-tuning for 100k steps per task in our method, collecting online interaction samples during the fine-tuning process. These samples were then incorporated as a supplementary dataset alongside the original data. Consequently, the resulting dataset expanded from 50M to 50M+100k×50. We subsequently trained the baseline methods on this augmented dataset so that the data used for our method and baseline methods are same. The experimental results, presented below, demonstrate that our method still outperform the baseline methods under this configuration. For MTDIFF, we utilize the default parameters provided by the authors. However, we observe a performance decline on these new datasets, which may be attributed to the increased inclusion of sub-optimal data during our online interaction phase.
>
> | Methods  |MT-50 Success rate|
> |----------|--------|
> | MTDIFF      | 27.06 ± 0.42|
> | HarmoDT      | 57.37 ± 0.34|
> | SODP      | 59.26 ± 0.18|
>
> **W3**: ... control experiments ...(1) fine-tuning with high-quality offline data and (2) direct training with high-quality data without pre-training.
>
> **A3**: Thank you for your insightful suggestions. We fine-tuned the models for 100k steps across five tasks and collected 200 successful episodes (equivalent to 100k steps) for each task. These datasets were subsequently utilized to train five models independently in an offline setting, using both our pre-trained model (SODP_off) and a model trained from scratch without pre-training (SODP_off scratch). We report the success rates across 3 seeds below. Without pre-training, the model lacks the action priors required to efficiently discover high-reward action distributions. Moreover, directly fine-tuning with high-quality offline data is inadequate, as static reward labels may not offer sufficient guidance for the model in dynamic environments and fail to facilitate efficient exploration.
>
> | Task                      |SODP_off|SODP_off scratch |SODP|
> |----------                 |--------|--------|--------|
> | button-press-topdown      |58.67 ± 0.03| 40.67 ± 0.08| 60.67 ± 0.03
> | hammer                    |71.33 ± 0.05| 13.33 ± 0.06| 73.33 ± 0.03
> | handle-pull-side          |60.67 ± 0.03| 42.67 ± 0.08| 81.67 ± 0.07
> | peg-insert-side           |25.33 ± 0.03| 0.0 ± 0.0   | 32.67 ± 0.06
> | handle-pull               |66.67 ± 0.03| 31.33 ± 0.06| 75.33 ± 0.04
> |Average success rate       |56.53 ± 0.18| 25.6 ± 0.18 | 64.73 ± 0.19

---

> > ### Author Response · Authors · 2024-11-26
> > **Response to Reviewer kFg1 (Part 2/2)**
> >
> > **W4**: A direct empirical comparison between optimal and sub-optimal pre-training data is missing.
> >
> > **A4**: Thank you for your insightful suggestions. To maintain a consistent total number of pre-training transitions, we select the final 50% of the converged SAC data, instead of the initial 50% (used as the sub-optimal dataset), as a near-optimal pre-training dataset for the Meta-World 10 tasks. Following the procedure outlined in our paper, we report the success rates across three seeds as follows. Incorporating a higher proportion of optimal transitions can enhance performance.
> >
> > | Task  |Sub-optimal dataset|Near-optimal dataset|
> > |----------|--------|--------|
> > | basketball          |52.67 ± 0.03| 80.67 ± 0.03|
> > | button-press        |88.00 ± 0.02| 89.33 ± 0.03|
> > | dial-turn           |80.67 ± 0.02| 74.00 ± 0.04|
> > | drawer-close        |100.00 ± 0.00| 100.00 ± 0.00|
> > | peg-insert-side     |62.67 ± 0.02| 84.67 ± 0.02|
> > | pick-place          |36.67 ± 0.03| 59.33 ± 0.03|
> > | push                |33.33 ± 0.03| 50.67 ± 0.03|
> > | reach               |68.67 ± 0.05| 95.33 ± 0.01|
> > | sweep-into          |60.67 ± 0.03| 75.33 ± 0.01|
> > | window-open         |69.33 ± 0.04| 100.0 ± 0.00|
> > | Average success rate|65.27 ± 0.21| 80.93 ± 0.16|
> >
> > **W5**: Why not directly BC in finetune phase.
> >
> > **A5**: Thank you for your insightful question. We conduct experiments using only Eq. (14) for fine-tuning, with the results presented below. Directly BC leads to poorer performance, as BC lacks reward labels to effectively guide exploration. This issue is akin to fine-tuning with offline high-quality data, as discussed in A3. Although BC during the fine-tuning phase can access dynamic actions, it is limited to 'imitation' rather than 'evolution,' since the model lacks the ability to differentiate between good and bad actions.
> >
> > | Task  |Directly BC|SODP|
> > |----------|--------|--------|
> > |button-press-topdown |51.3 ± 0.05| 60.7 ± 0.03|
> > | basketball          |21.3 ± 0.03| 41.2 ± 0.16|
> > | stick-pull          |26.7 ± 0.08| 50.5 ± 0.04|

---

### Official Review · Reviewer_4UTs · 2024-11-04

**Soundness:** 3
**Presentation:** 3
**Contribution:** 3
**Rating:** 5
**Confidence:** 3

**Summary:**

SODP is a two-stage diffusion-based framework designed to enhance multi-task planning in reinforcement learning. This framework comprises a pre-training phase that leverages large-scale, sub-optimal, task-agnostic data to learn a broad range of behaviors and a fine-tuning phase that uses task-specific rewards for adaptation. The pre-trained model captures general action patterns, allowing it to rapidly adapt to diverse tasks through fine-tuning, guided by reinforcement learning techniques. SODP demonstrates improved generalization in handling diverse tasks with minimal task-specific data and outperforms existing methods on benchmarks like Meta-World and Adroit.

**Strengths:**

* It leverages sub-optimal, task-agnostic data for pre-training, which contrasts with traditional reliance on task-specific demonstrations or expert data.
* The proposed behavior-cloning (BC) regularization in the fine-tuning stage offers a creative solution to the problem of model drift, allowing the model to maintain useful pre-trained capabilities while adapting to new tasks.

**Weaknesses:**

* Action and State Space Limitations: SODP requires tasks to share the same action space, limiting adaptability to tasks with different action spaces. It’s unclear how the method handles varying state space dimensions.

* Typographical Errors: There are minor errors, such as $a_0^t$ in Line 215, which should be $a^0_t$.

* Dependence on Online Fine-tuning: SODP relies on an online environment for fine-tuning, which may limit practical applications. The paper also lacks details on the number of fine-tuning steps and associated costs.

**Questions:**

* For different tasks with different state spaces, how to train these different tasks?
* After fine-tuning the pre-trained model on the specific task, can you evaluate the performance of the fine-tuned model on the other tasks? I'm very concerned that the catastrophic forgetting of the model has been addressed.
* Can you explain how to select the target policy in details?

---

> ### Author Response · Authors · 2024-11-26
> **Response to Reviewer 4UTs (Part 1/2)**
>
> Dear Reviewer 4UTs,
>
> We sincerely appreciate your precious time and constructive comments. In the following, we would like to answer your concerns separately.
>
> **W1**: Action and State Space Limitations: SODP requires tasks to share the same action space, limiting adaptability to tasks with different action spaces. It’s unclear how the method handles varying state space dimensions.
>
> **A1**: Thank you for your insightful questions. Following prior work, our goal is to enable a single agent to perform multiple tasks[1][2], so the action space for one agent is consistent (e.g., a 7-dimension action space for a 7-DOF robotic arm). However, recent research has explored cross-embodied robotic control, where the action spaces of different robotic arms may vary. These approaches typically address the discrepancy by mapping the distinct action spaces of each robot into a unified action space, effectively standardizing the action representation across different robots[3]. Then the different action space can also be viewed as one same action space.
> For variations in state dimensions, visual inputs can be handled straightforwardly by resizing images to a consistent format and processing them with a visual encoder. Similarly, low-dimensional state representations can be transformed into a unified state space, akin to the method used for harmonizing action spaces.
>
> **W2**: Typographical Errors: There are minor errors...
>
> **A2**: Thank you for your comment. We have corrected it and checked the entire manuscript again.
>
> **W3**: Dependence on Online Fine-tuning: SODP relies on an online environment for fine-tuning, which may limit practical applications. The paper also lacks details on the number of fine-tuning steps and associated costs.
>
> **A3**: Thank you for your insightful questions. Online interaction can indeed be costly and challenging, particularly in high-risk real-world environments where collecting sufficient trajectories is difficult. However, by leveraging existing offline data, a world model can be trained to simulate the transitions and feedback of real environments. Since our method only requires the judgment of actions planned by the model, it is well-suited for integration with a world model. Consequently, the entire learning process can occur within the model's simulated imagination, eliminating the need for direct access to the actual environments.
>
> In Meta-World, we fine-tune each task for $1e^6$ steps whcih requires about 15 hours on single A100 GPU. In Adroit, we fine-tune each task for $3e^3$ steps whcih requires about 3 hours on single A100 GPU.

---

> > ### Author Response · Authors · 2024-11-26
> > **Response to Reviewer 4UTs (Part 2/2)**
> >
> > **Q1**: For different tasks with different state spaces, how to train these different tasks?
> >
> > **A4**: See A1.
> >
> > **Q2**: After fine-tuning the pre-trained model on the specific task, can you evaluate the performance of the fine-tuned model on the other tasks? ...
> >
> > **A5**: Thank you for your insightful questions. We fine-tune three models on distinct tasks: *button-press-topdown (bpt)*, *button-press-topdown-wall (bptw)*, and *handle-pull-side (hps)*. These models are subsequently evaluated on 10 additional tasks. We report the evaluation suceess rate below and we observe that fine-tuning on a single task can enhance performance on some other tasks. This effect suggests that the fine-tuning process enables the model to acquire shared, transferable skills beneficial across multiple tasks without potential forgetting.
> > | Tasks                      |SODP_pretrain|SODP_bpt|SODP_bptw|SODP_hps|
> > |----------|--------|--------|--------|--------|
> > | button-press-topdown       |39.33±0.06|61.33±0.03|54.00±0.05|46.00±0.05
> > | button-press-topdown-wall  |43.33±0.05|57.33±0.01|60.67±0.04|46.00±0.02
> > | handle-pull-side           |56.67±0.05|52.67±0.02|46.00±0.09|88.67±0.01
> > | button-press               |32.33±0.07|47.33±0.06|44.67±0.03|43.33±0.08
> > | button-press-wall          |46.00±0.04|58.00±0.02|49.33±0.02|54.67±0.06
> > | coffee-button              |66.67±0.05|50.67±0.01|34.00±0.11|62.67±0.01
> > | dial-turn                  |44.33±0.06|34.67±0.08|28.67±0.03|26.00±0.01
> > | door-close                 |80.00±0.09|86.00±0.07|86.67±0.08|82.00±0.07
> > | door-lock                  |20.67±0.04|17.33±0.03|24.33±0.09|27.33±0.04
> > | door-open                  |60.67±0.04|66.67±0.03|72.00±0.02|64.67±0.04
> > | drawer-close               |68.67±0.06|76.00±0.06|91.00±0.04|87.00±0.04
> > | Averagte success rate      |50.78±0.06|55.27±0.05|53.76±0.06|57.12±0.07
> >
> > **Q3**: Can you explain how to select the target policy in details?
> >
> > **A6**: We define the target policy as the expert policy capable of successfully solving the specific downstream task. To approximate this policy, we maintain an expert replay buffer during the fine-tuning stage, storing all successful online interactions along with their corresponding denoising latents.
> >
> > **References**
> >
> > [1] He et al. "Diffusion model is an effective planner and data synthesizer for multi-task reinforcement learning." In NeurIPS 2024.
> >
> > [2] Hu et al. "HarmoDT: Harmony Multi-Task Decision Transformer for Offline Reinforcement Learning." In ICML 2024.
> >
> > [3] Liu et al. "RDT-1B: a Diffusion Foundation Model for Bimanual Manipulation." arXiv 2024.

---

### Official Review · Reviewer_NpAc · 2024-11-04

**Soundness:** 3
**Presentation:** 3
**Contribution:** 3
**Rating:** 5
**Confidence:** 3

**Summary:**

This paper introduces a novel framework called SODP, designed to train a multi-task diffusion planner using sub-optimal data. SODP aims to reduce dependency on task-specific labeled data through a two-stage training process. Initially, a diffusion model is pre-trained without the need for task-specific rewards or demonstrations. Subsequently, the model undergoes a second stage of online interactive reinforcement learning-based fine-tuning to rapidly refine its capabilities. The effectiveness of this methodology is validated on two multi-task domains: Meta-World and Adroit. Experimental results show that SODP outperforms the considered baseline methods, highlighting its potential in multi-task learning environments.

**Strengths:**

1. The paper is well-written and includes illustrative figures that aid comprehension.

2. It reports that SODP not only surpasses baseline methods in performance but also shows rapid convergence and robustness across diverse tasks and input modalities. Additionally, the included ablation study offers valuable insights into the design and effectiveness of the proposed methods.

**Weaknesses:**

My primary concerns arise from the evaluation protocols, specifically:

1.	Was the fine-tuning stage also applied to the baseline methods? If not, this might represent an unfair comparison, as all baselines are trained on sub-optimal data while SODP gains an advantage from an additional online fine-tuning stage.
2.	Regarding the Adroit task, was the 3D visual feature process (from DP3) also employed for the Diffusion Policy in Table 2? Moreover, what is the performance of SODP when this process is not utilized?

**Questions:**

1. Is the 3D visual feature process not compatible with MTDIFF? Have the authors tried combining the 3D visual feature process with MTDIFF on the Adroit task?
2. What does the vertical orange dashed line indicating in Figure 4?

---

> ### Author Response · Authors · 2024-11-26
> **Response to Reviewer NpAc**
>
> Dear Reviewer NpAc,
>
> We sincerely appreciate your precious time and constructive comments. In the following, we would like to answer your concerns separately.
>
> **W1**: Was the fine-tuning stage also applied to the baseline methods? ... this might represent an unfair comparison...
>
> **A1**: Since the baseline methods (MTDIFF and HarmoDT) are offline algorithms, we did not apply them to online interaction settings. To ensure a fair comparison, we conducted fine-tuning for 100k steps per task in our method, collecting online interaction samples during the fine-tuning process. These samples were then incorporated as a supplementary dataset alongside the original data. Consequently, the resulting dataset expanded from 50M to 50M+100k×50. We subsequently trained the baseline methods on this augmented dataset so that the data used for our method and baseline methods are same. The experimental results, presented below, demonstrate that our method still outperform the baseline methods under this configuration. For MTDIFF, we utilize the default parameters provided by the authors. However, we observe a performance decline on these new datasets, which may be attributed to the increased inclusion of sub-optimal data during our online interaction phase.
>
> | Methods  |MT-50 Success rate|
> |----------|--------|
> | MTDIFF   | 27.06 ± 0.42|
> | HarmoDT  | 57.37 ± 0.34|
> | SODP     | 59.26 ± 0.18|
>
> **W2**: Regarding the Adroit task, was the 3D visual feature process (from DP3) also employed for the Diffusion Policy in Table 2? Moreover, what is the performance of SODP when this process is not utilized?
>
> **A2**:
> - DP3 is the variant of Diffusion Policy that incorporates the 3D visual feature processing module, whereas the original Diffusion Policy employs a ResNet18 encoder for visual feature extraction.
> - We replace the DP3 encoder with the same visual encoder used in original Diffusion Policy and conduct experiments on Adroit 3 tasks. The experimental results are shown below. Our methods still outperform Diffusion Policy using original image inputs.
>
> | Methods  |Hammer|Door|Pen|Average
> |----------|--------|--------|--------|--------|
> | Diffusion Policy   | 36 ± 11| 65 ± 3| 18 ± 4| 39.8
> | SODP_img           | 52 ± 5| 90 ± 2| 40 ± 3| 60.1
> | SODP_3D            | 67 ± 6| 96 ± 1| 59 ± 4| 73.9
>
> **Q1**: Is the 3D visual feature process not compatible with MTDIFF? Have the authors tried combining the 3D visual feature process with MTDIFF on the Adroit task?
>
> **A3**: The 3D visual feature processing module can be integrated into MTDIFF. However, due to the incompatibility of MTDIFF with the different action spaces of different tasks in Adroit, we applied this integration to the image-based Meta-World 10 task instead. The experimental results, shown below, further validate the effectiveness of our method in complex high-dimentional scenarios.
> | Methods  |Image-based Meta-World 10 success rate|
> |----------|--------|
> | DP3        | 32.6 ± 0.23|
> | MTDIFF_3D  | 38.0 ± 0.82|
> | SODP       | 47.5 ± 0.18|
>
> **Q2**: What does the vertical orange dashed line indicating in Figure 4?
>
> **A4**：The vertical orange dashed line indicates the completion of our fine-tuning process. Our method requires approximately 0.65 million gradient steps in total, encompassing both the pre-training and fine-tuning stages, whereas the baseline methods necessitate 1.5 million gradient steps for training.

---

> > ### Comment · Reviewer_NpAc · 2024-12-03
> >
> > Thank the authors for the detailed explanation. Regarding W1, I have a concern about whether this comparison is fair. Fine-tuning online could correct biases that the model learned during the offline stage. However, using an offline dataset collected from SODP does not seem to provide an opportunity for the model to self-correct through trial and error.

---

### Meta-Review · Area_Chair_tv3W · 2024-12-21

**Metareview:**

The paper introduces SODP, a two-stage framework for robotic manipulation tasks, which utilizes suboptimal data to train a Diffusion Planner. The approach pre-trains a diffusion model on sub-optimal multi-task data and fine-tunes it using policy gradient optimization with a novel BC regularization. The framework leverages sub-optimal data to avoid reliance on high-quality labels while achieving strong performance during fine-tuning.

Reasons to accept
- The paper is well-structured and easy to follow.
- The proposed BC regularization effectively balances retaining pre-trained knowledge and exploring new high-reward behaviors, supported by thorough ablations.
- The paper presents a novel method to leverage sub-optimal data.
- The proposed method achieves strong empirical performance on Meta-World and Adroit

Reasons to reject
- The role of pre-training with sub-optimal data is not rigorously isolated, and critical experiments (e.g., pre-training with high-reward data or direct fine-tuning without pre-training) are missing.
- Experimental results omit the computational costs of pre-training (e.g., 500k steps) and lack clarity on whether baselines also benefit from online data collection, potentially biasing comparisons.
- The absence of comparisons with straightforward methods, such as direct behavior cloning during fine-tuning, weakens the argument for the necessity of the proposed framework.
- The performance gains might primarily stem from BC regularization rather than the fine-tuning framework, which undermines the novelty of the approach.

While this paper studies a promising research direction and presents an interesting approach, I believe its weaknesses outweigh its strengths. Consequently, I recommend rejecting the paper.

**Additional Comments On Reviewer Discussion:**

During the rebuttal period, 3 reviewers acknowledged the author's rebuttal, and 2 reviewers adjusted the score accordingly.

---

### Decision · Program_Chairs · 2025-01-22

Reject